# AlphaPept: a modern and open framework for MS-based proteomics

Maximilian T. Strauss ®[1,2] ✉, Isabell Bludau ®[1], Wen-Feng Zeng ®[1], Eugenia Voytik[1], Constantin Ammar[1], Julia P. Schessner ®[1], Rajesh Ilango[3], Michelle Gill[3], Florian Meier ®[1,4], Sander Willems[1] & Matthias Mann ®[1,2] ✉

In common with other omics technologies, mass spectrometry (MS)-based proteomics produces ever-increasing amounts of raw data, making efficient analysis a principal challenge. A plethora of different computational tools can process the MS data to derive peptide and protein identification and quantification. However, during the last years there has been dramatic progress in computer science, including collaboration tools that have transformed research and industry. To leverage these advances, we develop AlphaPept, a Python-based open-source framework for efficient processing of large high-resolution MS data sets. Numba for just-in-time compilation on CPU and GPU achieves hundred-fold speed improvements. AlphaPept uses the Python scientific stack of highly optimized packages, reducing the code base to domain-specific tasks while accessing the latest advances. We provide an easy on-ramp for community contributions through the concept of literate programming, implemented in Jupyter Notebooks. Large datasets can rapidly be processed as shown by the analysis of hundreds of proteomes in minutes per file, many-fold faster than acquisition. AlphaPept can be used to build automated processing pipelines with web-serving functionality and compatibility with downstream analysis tools. It provides easy access via one-click installation, a modular Python library for advanced users, and via an open GitHub repository for developers.

Increasingly large datasets, combined with likewise increasing computational power and algorithmic advances are transforming every aspect of society. This is accompanied and enabled by developments in open and transparent science. The open-source community has been a particular success, starting as a fringe movement to become a recognized standard for software development whose value is embraced and adapted even by the largest technology companies. Public exposure supports high code quality through scrutiny by developers from diverse backgrounds, while increasingly sophisticated collaboration mechanisms allow rapid and robust development cycles. The most advanced machine and deep learning research, for

example, builds on open-source projects and datasets and is itself open-source. These laudable developments reflect the core ideas of science and present great opportunities in the ever more important computational fields.

In mass spectrometry (MS)-based proteomics, algorithms and computational frameworks have been a cornerstone in interpreting the data, resulting in a large variety of different proteomic software packages and algorithms, ranging from commercial, and freely available to open-source, exemplified by and reviewed in ref. 1,2. Typical computational workflows comprise the detection of chromatographic features, peptide-spectrum matching, all the way through protein

[1]Department of Proteomics and Signal Transduction, Max Planck Institute of Biochemistry, Martinsried, Germany. [2]NNF Center for Protein Research, Faculty of Health Sciences, University of Copenhagen, Copenhagen, Denmark. [3]Nvidia Corporation, Santa Clara, CA, USA. [4]Functional Proteomics, Jena University Hospital, Jena, Germany. ✉e-mail: maximilian.strauss@cpr.ku.dk; mmann@biochem.mpg.de

inference and quantification[3,4]. Advances in (MS)-based proteomics are also being accelerated through the sharing of datasets, such as publicly available data on the Proteome Exchange repository[5,6] and the application of deep learning technologies[7,8]. There are increased efforts to achieve reproducible computational workflows by using pipelines such as the Nextflow framework[9] that allow scalable execution using software containers. Community efforts such as nf-core[10] build on Nextflow for collaborative, peer-reviewed, best-practice pipelines, such as quantms for proteomics[11].

Note that this array of tools includes widely used software with innovative and performant algorithms. Yet, closed-source software and lack of transparency and continuous testing can limit the potential for wider scientific collaboration and progress. We believe that ultimately code, being a method, should be published as part of the results themselves. Developments in many areas, not least in AI, underscore the immense value of open-source alternatives. Not only does this foster a culture of collaboration and continuous refinement, but it ultimately encourages trust in scientific results.

Prompted by the developments in the Python scientific environment and in collaborative development tools, we developed Alpha-Pept, a Python-based open-source framework for the efficient processing of large amounts of high-resolution MS data. Our main design goals were accessibility, analysis speed, and robustness of the code and the results. Accessibility refers to the idea of facilitating the contribution of algorithmic ideas for (MS)-based proteomics, which is today typically limited to bioinformatics experts. We decided on Python because of its clear, easy-to-understand syntax and because the excellent supporting scientific libraries make it easier for developers from different backgrounds to contribute to and implement innovative ideas. Using community-tested packages makes the code base more maintainable and robust, allowing us to focus on domain knowledge instead of implementation details. We furthermore adopted a recent implementation of "literate programming"[12], in which code and documentation are intertwined. Using the nbdev package, the code base is connected to extensive documentation in Jupyter Notebooks in a way that immediately explains the algorithmic background, making it easier to understand the underlying principles and document design decisions for others[13]. With the help of the Numba package for just-in-time compilation (JIT) of Python code[14], AlphaPept achieves very fast computation times. Furthermore, we implemented

robust design principles of software engineering on GitHub, such as continuous integration, deployment and extensive automated validation.

Depending on the user, AlphaPept can be employed in multiple ways. A "one-click" installer can be freely downloaded for Windows, provid†ing a web server-based graphical user interface (GUI) and a command-line interface; A Python library that allows re-use and modification of its functionality in custom code, including in Jupyter Notebooks that have become a standard in data science and finally, in a scalable could environment.

In the remainder of the paper, we describe the functionality of AlphaPept on the basis of nbdev notebooks, such as feature finding, peptide identification and protein quantification. We demonstrate the capabilities of AlphaPept on small- and large-scale datasets. Finally, we demonstrate how AlphaPept can be utilized as a proteomic workflow management system and how it can be integrated with downstream analysis tools such as Perseus or the Clinical Knowledge Graph (CKG)[15,16], and provide an outlook on further developments.

## Results
### Overview of AlphaPept architecture
Academic software development is often highly innovative but is rarely undertaken with dedicated funding or long-term personnel stability. Such constraints have successfully been mitigated by collaborative software engineering approaches and the collective efforts of volunteers. This is exemplified in state-of-the-art open-source projects such as NumPy[17] and scikit-learn[18]. This paradigm has also been taken over by relatively recent and highly popular deep learning frameworks like Google's Tensorflow[19] and Facebook's PyTorch[20] and is thought to lead to increased code quality due to community exposure and a large testing audience. Inspired by these developments, AlphaPept implements robust design principles of software engineering on GitHub, such as continuous testing and integration. For instance, code contributions can be made via pull requests which are automatically validated. By making the code publicly available and providing a stringent testing environment, we hope to encourage contribution and testing from a diverse background while maintaining very high code quality. In the realm of proteomics, the recent open-source example of Sage[21], which uses the same fragment-ion indexing as the closed-source

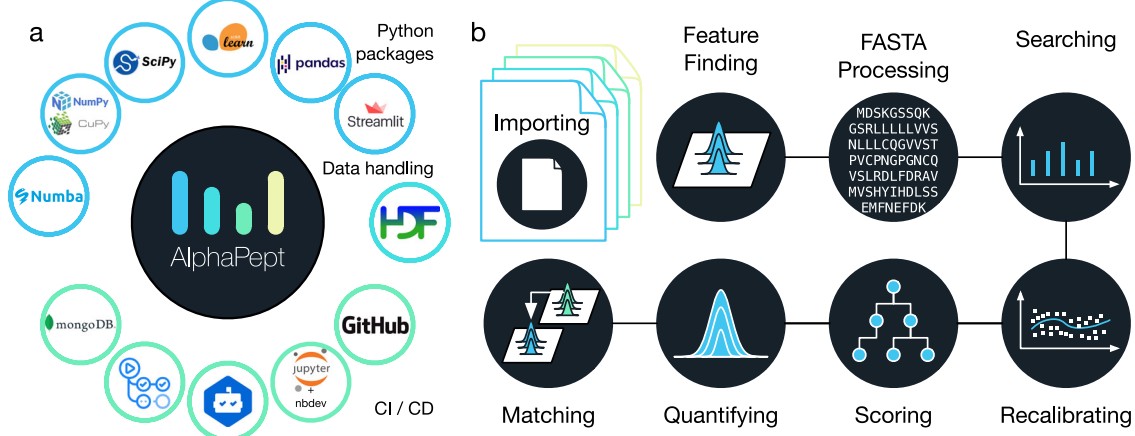

**Fig. 1 | AlphaPept "ecosystem" and modules. a** AlphaPept relies on multiple community-tested packages. We use highly optimized libraries such as Numba, NumPy, CuPy, scikit-learn, SciPy, and pandas to achieve performant code. As GUI, we provide a browser-based application built on streamlit. For data handling, the HDF5 file technology is used. The repository itself is hosted on GitHub, and the core code is documented in Jupyter Notebooks using the nbdev package. To ensure maintainability, packages are continuously monitored for updates via dependabot. New code is automatically validated using GitHub actions and summary statistics (timing, identifications, and quantifications) are uploaded to a MongoDB database and visualized. **b** All algorithmic code of AlphaPept is organized in Jupyter Notebooks. For the key processing steps in the pipeline, such as importing raw data, Feature Finding, FASTA processing, Searching, Recalibrating, Scoring, Quantifying, and Matching, there are individual notebooks with background information and the code.

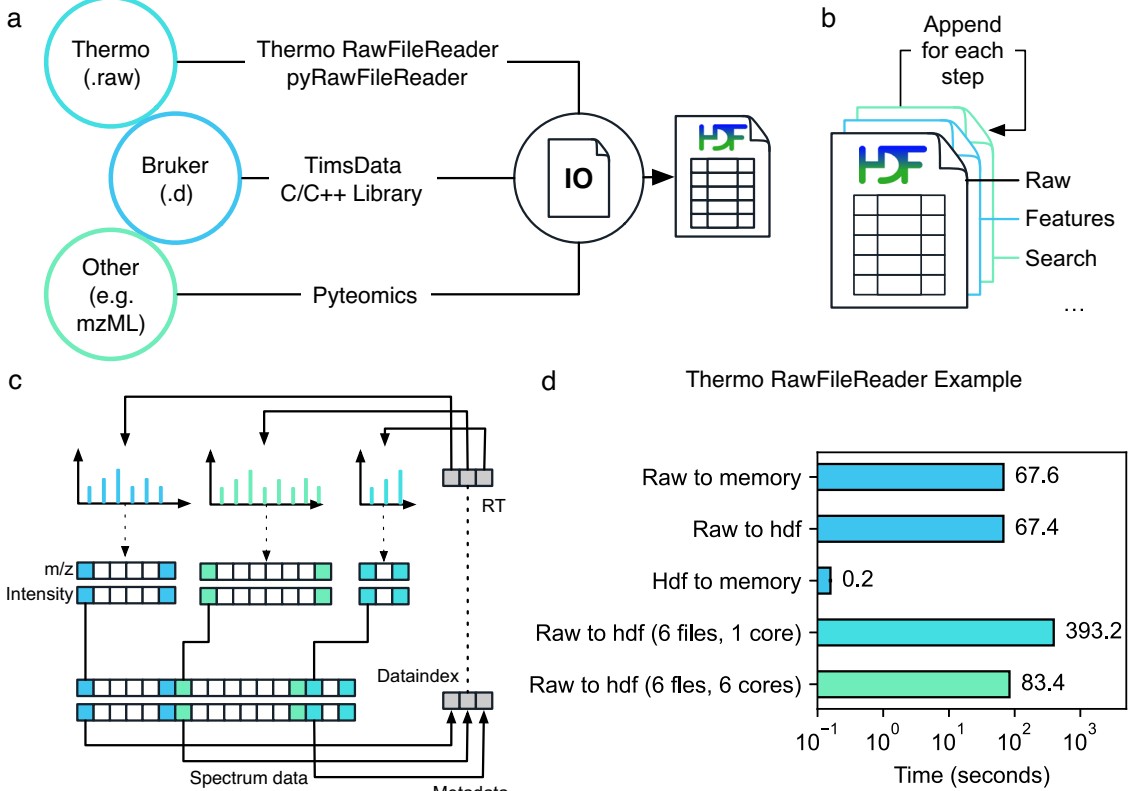

**Fig. 2 | Highly efficient and platform-independent MS data access. a** MS data from different vendors is imported into an HDF5 container for fast and platform-independent data access. To read Thermo data, we provide a Python application programming interface. Bruker data is accessed via Bruker's proprietary DLL. Additionally, generic data can be imported using the Pyteomics package. **b** The output of each processing step is appended to the HDF5, allowing processing in a modular way. **c** To efficiently store MS spectra, multiple spectra of variable length are concatenated, and start indices are saved in a lookup table. **d** HDF5 Accessing times. Loading data from HDF5 into memory takes less than 1 s for a typical 2 h full proteome analysis of a HeLa sample acquired on a Thermo Orbitrap mass spectrometer.

MSFragger[22], exemplifies how performant open-source software is cherished by the community.

Organization in notebooks with nbdev allows us to collect documentation, code, and tests in one place. This enables us to automatically generate the documentation, extract production code and test functionality by executing the notebooks. From a user perspective, this can serve as an ideal educational resource as changes in the code can be directly tested. Furthermore, we extend the notion of unit and system testing by including real-world datasets on which the overall improvement of newly implemented functionality is routinely evaluated. To continuously monitor system performance, summary statistics are automatically uploaded to a database where they are visualized in a dashboard.

The advantages of high-level languages generally come at the price of execution speed, especially for Python. As a result, this expressive language is often only used as a thin wrapper on C++ libraries. In AlphaPept, we make use of the Numba project[14], which allows us to compile our Python algorithms directly with the industry-standard LLVM compiler (backend to most C++ compilers and supercomputing languages such as Julia). This allows us to speed up our code by orders of magnitude without losing the benefits of the intuitive Python syntax. We note that while Numba functions are written in Python, they need to be optimized for the task at hand, additionally emphasizing the need for transparent code. The current AlphaPept code base utilizes design patterns that were only possible with later updates of the Numba package. Furthermore, AlphaPept readily parallelizes computationally intensive parts of the underlying algorithms

on multiple CPU cores or—if available—Graphical Processor Units (GPUs) for further performance gains.

As far as possible, AlphaPept uses the standard, but powerful packages of the Python data analysis universe, namely NumPy for numerical calculations, pandas[23] for spreadsheet-like data structures, CuPy[24] for GPU, and scikit-learn for machine learning (ML) (Fig. 1a). Furthermore, we chose the binary, high-performance HDF5 (hierarchical data format 5) file format, which is used across scientific areas, including 'big data' projects (see below). All these packages are platform-independent, allowing deployment of AlphaPept on Windows, Mac, and Linux computers, including cloud environments.

An integral feature of AlphaPept development are Jupyter notebooks, which have become ubiquitous in scientific computing. Using the nbdev package, each part of the MS-based proteomics workflow is modularized into a separate notebook. This allows extensive documentation of the underlying algorithmic production code, which is automatically extracted from and synchronized with the notebooks. Furthermore, the notebooks capture the background information of each part of the computational proteomics workflow, making it much easier to understand the underlying principles. As Jupyter Notebooks can be interactively explored and each code cell can be individually executed, this serves as an educational resource because algorithmic changes can be readily tested on a cell-level within the Notebook. We have found this to be an excellent way of developing software, which brings together the typical cycle of exploration in notebooks with the production of a robust and tested code base. Figure 1b shows an overview of the steps in the analysis of a typical proteomics

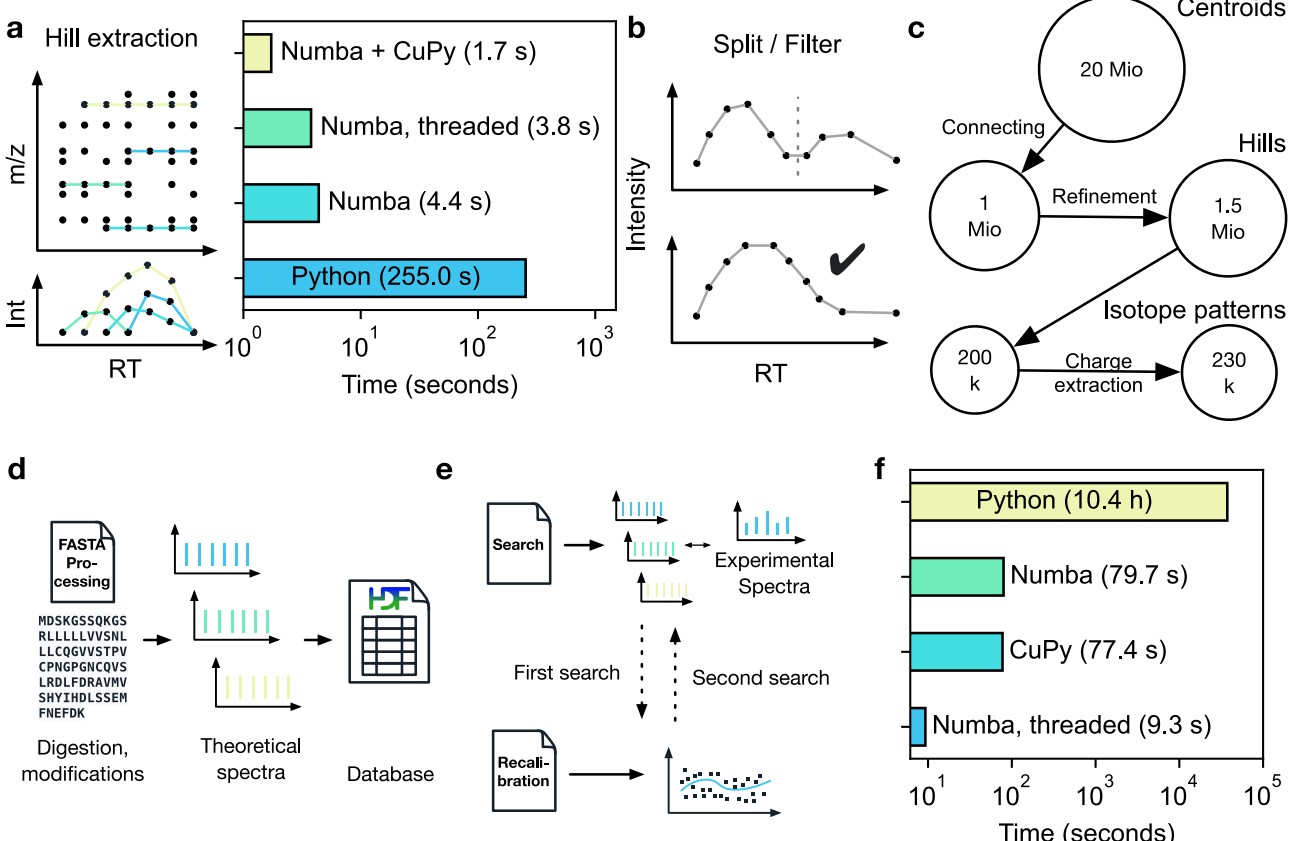

**Fig. 3 | Extracting isotope features and database search. a** Individual MS peaks of similar masses are connected over the retention time using a graph approach, resulting in "hills". Using a native Python implementation, hill extraction takes several minutes. Numba, parallelization on CPUs or GPUs reduces hill extraction to seconds. CuPy refers to using GPU, Numba for single-threaded implementation, and Numba threaded to Numba using multiple threads. **b** Extracted hills are refined by splitting at local minima and only allowing well-formed elution profiles. **c** Starting with 20 million points for a typical Thermo HeLa shotgun proteomics file, these are connected to approximately one million hills, which increased to 1.5 million after hill splitting and filtering. Subsequent processing results in 200,000 pre-isotope patterns that ultimately yield 230,000 isotope patterns due to assignment to specific charge states. **d** The FASTA processing notebook contains functionality to calculate fragment masses from FASTA files which are saved in an HDF5 container for subsequent searches. **e** Initially, a first search is performed, and masses are subsequently recalibrated. Based on this recalibration, a second search with more stringent boundaries is performed. **f** Using the decorator strategy, the search can be drastically sped up, from 10 h in a pure Python implementation to seconds with Numba and CuPy.

experiment in AlphaPept corresponding to the notebooks. These separate processing steps will be discussed in turn in the sections below.

## Highly efficient and platform-independent MS data access

MS-based proteomics or metabolomics generates complex data types of MS1-level features, variable length MS2 data, and mappings between them. Furthermore, data production rates are rapidly increasing, making robust and fast access a central requirement. The different MS vendors have their own file formats, which may be highly optimized but are meant to be accessed by their own software and are, in some cases, confined to certain operating systems. We therefore faced the task of extracting the raw data into an equally efficient but vendor and operating system-neutral format that could be accessed rapidly.

First, AlphaPept needs to convert vendor-specific raw files. For Thermo files, we created a cross-platform Python application programming interface (API) that can directly read .RAW MS data (pyRawFileReader, Fig. 2a). It uses PythonNET for accessing Thermo's RawFileReader.NET library[25], obviating the need for Thermo's proprietary MSFileReader. For Windows, PythonNET can be directly installed with Windows'.NET Framework. For Linux and MacOS, PythonNET requires the open-source Mono library. Although our solution uses stacked APIs, loading the spectra of a Thermo.RAW file of 1.6 Gb into RAM takes only about one minute, which can be speeded up

even more by parallel file processing. Access to Bruker's timsTOF raw data is also directly handled from our Python code, in this case through a wrapper to the external timsdata.dll C/C++ library, both made available by Bruker. In parallel with this publication, we provide AlphaTims[26], a highly efficient package to access large ion mobility time-of-flight data through Python slicing syntax and with ultra-fast access times (https://github.com/MannLabs/alphatims). While this ensures cross-platform compatibility, we recommend file conversion on an operating system similar to the acquisition computer, due to vendor-imposed dependencies to read the files.

To accommodate raw data acquired through other vendors, we use Pyteomics[27,28]. This package allows reading mzML and other standard MS data formats with Python. Thus, by first converting raw data with external software such as, e.g., MSConvert[29], AlphaPept also provides a generic framework for all vendors.

As a storage technology, we chose HDF5 (Hierarchical Data Format 5), a standard originally developed for synchrotron and other large-scale experimental datasets, that has now become popular in a wide range of scientific fields[30]. HDF5 has many benefits, such as independence of operating systems, arbitrary file size, fast accession, and a transparent, flexible data structure. The latter is achieved by organizing HDF5 files in groups and subgroups, each containing arrays of arbitrary size and metadata which describes these arrays and (sub) groups. Multiple community tools allow the exploration of HDF5 files,

further facilitating user access to the raw data. It is also becoming more popular in the field of MS[31]. AlphaPept adopts the HDF5 technology via Python's h5py package[32].

As an additional design choice, we also store intermediate processing results in the HDF5 container so that individual processing steps can be performed modularly and from different computers. This enables researchers to quickly implement and validate new ideas within the downstream processing pipeline. Thus, for each new sample, AlphaPept creates a new.ms_data.hdf file and for each step in the workflow, the file is extended by a new group (Fig. 2b). In this way, the.ms_data.hdf file ensures full portability transparency and reproducibility while being fast to access and with minimal storage requirements. For example, the 1.6 Gb Thermo file mentioned above is converted to an HDF5 file of 200 MB, all of which can be accessed in a total of 0.2 s (Fig. 2d).

We next provide functionality for MS data pre-processing, such as centroiding and extraction of the n-most abundant fragments, should this not already have happened in the vendor software. MS1 and MS2 scans form the two major subgroups in the HDF5 file. As HDF5 files are not optimized for lists of arrays with variable lengths, we convert the many individual spectra into a defined number of arrays, each containing a single data type, but concatenating all spectra. These arrays are organized in two sets: Spectrum metadata (spectrum number, precursor m/z, RT, etc.), where each array position corresponds to one spectrum; and spectrum data, where each array position corresponds to a single m/z-intensity pair. To unambiguously match the spectrum datapoints to their metadata, an index array is created. It is part of the first set of arrays and contains a pointer to the position of the first data pair for each spectrum within the second set. The position of the last pair does not need to be stored as it is implied by the start position of the next spectrum. Thereby, all m/z values and intensities for each spectrum can easily be extracted with simple base Python slicing, while fixing the number of arrays contained in the hdf container. Loading data from HDF5 to RAM takes less than a second, effectively speeding up data accession more than 300-fold compared to loading the RAW file (Fig. 2d).

### Extracting isotope features

Having stored the MS peaks from all mass spectra in an efficient data structure, we next determine isotope patterns over chromatographic elution profiles. This computationally intensive task is crucial for subsequent peptide identification and quantification. MaxQuant[33] introduced the use of graphs for feature finding, which was then improved upon by the Dinosaur tools[34], and we also decided to follow this elegant approach. Another recent development that builds on Dinosaur in Python is Biosaur[35].

In the first step—called hill building—centroided peaks from adjacent scans are connected. As there are millions of centroids, our first implementations using pure Python took several minutes of computing time. We subsequently refactored the graph problem and parallelized it for CPUs using Numba and CuPy for GPUs, resulting in a 150-fold speed up (about 2 s on GPU). Since not every user has access to GPUs, AlphaPept employs dedicated Python "decorators", a metaprogramming technique allowing a part of the program to modify its another part at compile time to transparently switch between parallelized CPU, GPU, and pure Python operation.

In more detail, AlphaPept refines hills by first splitting them in case they have local minima indicating two chromatographic elution peaks (Fig. 3a). Additionally, hills are removed whose elution profiles do not conform to minimal criteria, like minimal length and the existence of local minima. To efficiently connect hills, we compute summary statistics such as weighted average m/z value and a bootstrap estimate of its precision. Hills within retention time boundaries are grouped into pre-isotope patterns. To correctly separate co-eluting features, we generate seeds, which we extend in elution time and check

for consistency with a given charge state, similarity in elution profile, and conformity with peptide isotope abundance properties via the averagine model[36]. This results in a feature (here a possible peptide precursor mass), which is described by a table.

Feature finding on the Bruker timsTOF involves ion mobility as an additional dimension. Currently, this functionality is provided by a Bruker component, which we linked into our workflow via a Python wrapper, and is the only part that is not in natively included as Python code in AlphaPept. Instead, this wrapper uses Python's subprocess module, which can integrate other tools into AlphaPept just as easily.

For a typical proteomics experiment performed on an Orbitrap instrument, Fig. 3c provides an overview of the number of datapoints from MS peaks to the final list of isotope patterns. Note that AlphaPept can perform feature finding separately for each file as soon as it is acquired (described below). Furthermore, although described here for MS1 precursors, the AlphaPept feature finder is equally suited to MS2 data that occur in parallel reaction monitoring (PRM) or DIA (data-independent acquisition) acquisition modes.

### Peptide-spectrum matching

The heart of a proteomics search engine is the matching of MS2 spectra to peptides in a protein sequence database. AlphaPept parses FASTA files containing protein sequences and descriptions, "digests" them into peptides, and calculates fragment masses according to user-specified rules and amino acid modifications (Fig. 3a). We again use HDF5 files, which enables efficient storage of fragment series despite their varying lengths. Generation of this database only happens once per project and only takes minutes for typical organisms and modifications. From a FASTA file of the human proteome, typically, five million "in silico" spectra of fragment masses are generated. In case no enzyme cleavage rules are specified or for open search with wide precursor mass tolerances, the fragments are instead generated on the fly to avoid excessive file sizes.

To achieve maximum speed, AlphaPept employs a very rapid fragment-counting step to determine initial peptide-spectrum matches (PSMs). In brief, this step involves two pointers that iterate over the lists of sorted theoretical and experimental fragment masses and compare their mass difference. If the mass difference between a fragment pair falls within the search tolerance, this is considered a hit, and the counter is incremented by one. As this step only involves the addition and subtraction of elements in numerical arrays, the machine code produced by Numba is very efficient and easily parallelized. For each hit, we additionally compute the fraction of MS2 signals accounted for by the match and add this fraction to the counter. Consequently, for each peptide-spectrum match, we store a floating-point number that represents the integer number of hits and the matched intensity fraction of these hits, effectively re-implementing the Morpheus score. This score, despite its computational simplicity, can outperform the somewhat more complex X! Tandem score[37]. This leaves a much smaller number of peptides that have at least a minimum number of fragment matches to the experimental spectrum. For the human proteome and mass measurement accuracy of parts per million, the initial millions of comparisons are decreased to a maximum of Top-N remaining candidates per MS2 spectrum (typically 10). Supplementary Fig. 1 provides a comprehensive overview of how the number of identified precursors changes with varying Top-N depending on the score used. This enables more computationally expensive scoring in the second step. Different scores can be implemented in AlphaPept, such as the widely used X!Tandem score[38]. We have implemented a 'generic score', that is inspired by the Morpheus score (number of hits and matched intensity fraction), the intuition behind the X!tandem hypergeometric distribution, and the length of the matched sequence (See Supplementary Formula 1). Note that the sole function of this score is to rank the PSMs, whereas statistical significance is

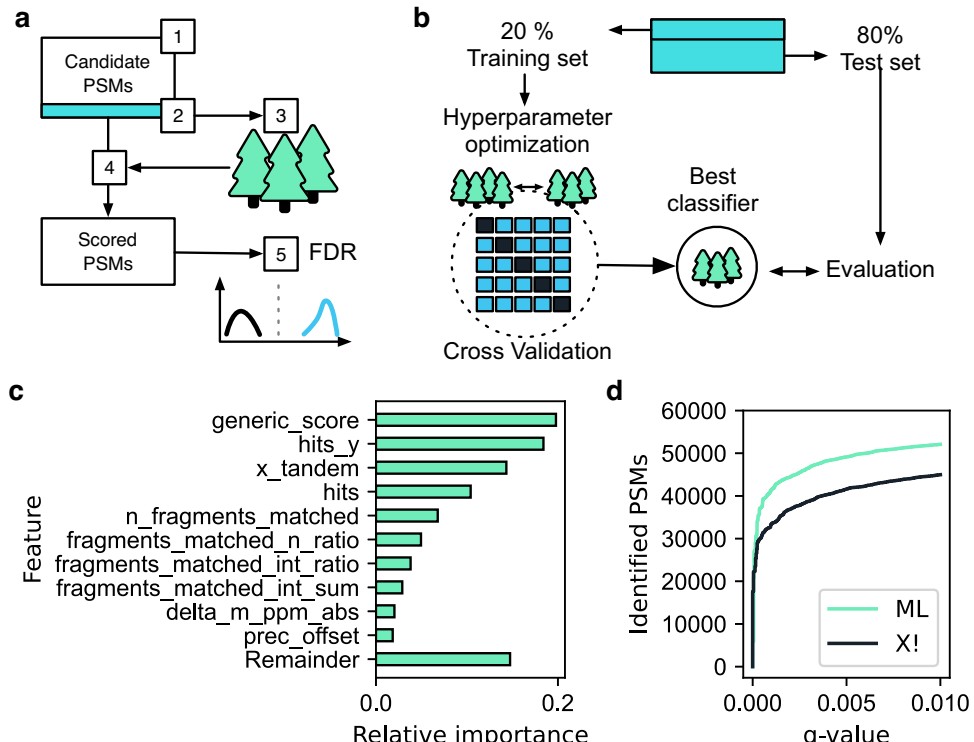

**Fig. 4 | Machine learning-based scoring and FDR estimation. a** We train a Random Forest (RF) classifier on a subset of candidate PSMs to distinguish targets from decoys based on PSMs characteristics. A semi-supervised machine learning model is applied with the following steps: (1) extraction of all candidate PSM scores, (2) selection of a PSM subset for machine learning, (3) training of an RF classifier, and (4) application of the trained classifier to the full set of PSM candidates. Finally, the probability of the RF prediction is used as a score for subsequent FDR control (5). **b** Training of the classifier (step 4 in panel **a**) follows a train-test split scheme where only a fraction of the candidate subset is used for training. Using stringent cross-validation, multiple hyperparameters are tested to achieve optimal RF performance. The best classifier is benchmarked against the remaining test set. **c** Example feature importance for an Orbitrap test set, where the number of y-ion hits is the highest contributing factor to the model. Note that the RF algorithm can utilize any database identification score, such as the X!Tandem score chosen here, which is the fourth most important feature. The generic_score, our "generic score", is a score based on the peptide length, the total number of fragment hits, b-ion hits, and the matched intensity ratio. See the *AlphaPept workflow and files* Notebook for an explanation of features. **d** Optimized identification with the ML score. Compared to the X!Tandem score alone, the ML optimization identified about 14.4% more PSMs for the same $q$ value.

determined by counting reverse database hits and by machine learning (see below).

We perform a first search for the purpose of recalibrating the mass scale as a function of elution time (Fig. 3e). Here, we use weighted nearest neighbor regression instead of binning by retention time (explained in the accompanying Jupyter Notebook). The k-nearest neighbors regressor that we selected allows non-linear grouping in several dimensions simultaneously (retention time and mass scale in the case of Orbitrap data and additionally ion mobility in the case of timsTOF data).

Having recalibrated the data, the main search is performed with an adapted precursor tolerance. We furthermore calculate the matched ion intensity, matched ions, and neutral loss matches for further use and reporting together with charge, retention time, and other data.

To demonstrate the speed-up achieved by our architecture and the performance decorator, we timed illustrative examples (Fig. 3f). On a HeLa cell line proteome acquired in a single run, comparing 260k spectra to 5 million database entries, the computing time in pure Python was about 10 h. This decreased to 80 s when employing Numba (~450x improvement), to 77 s when using Numba with CuPy on GPU, and further to 9 s on multi-threaded CPU (see companion Figure Notebook). The GPU acceleration is not larger because the code is already very efficient on the CPU, and some workflow tasks are memory-bound instead of computationally bound. Improved memory management on GPU could further decrease GPU computational time. In any case, AlphaPept reduces the PSM matching step to an insignificant part total computation time. Note that GPUs can achieve higher

processing speeds over single CPUs and are therefore particularly useful for specialized operations, for instance, real-time. However, we found that for runs with default settings and especially when processing multiple files in parallel, CPUs with multiple processors are usually better suited.

## Machine learning-based scoring and FDR estimation

Assessing the confidence of PSMs requires a scoring metric that separates true (correctly identified) from false (wrongly identified) targets in the database. Multiple defined features are calculated by the AlphaPept search engine and used in a score to rank the targets. A nonsense database of pseudo-reversed sequences where the terminal amino acid remains unchanged[39] is used to directly estimate the False Discovery Rate (FDR) by counting reverse hits. Score thresholds subsequently decide which targets should be considered identified. To further validate this approach and to ensure accurate FDR estimation across different development stages in AlphaPept, our GitHub testing routine includes an empirical two-species FDR test based on an "entrapment strategy"[40–42].

In recent years, machine learning has gained increasing momentum in science in general, but also in its specific applications to MS data analysis. One of the first of these was the combination of multiple scoring metrics to a combined discriminant score that best separates high-scoring targets from decoys. This was initially integrated into PSM scoring through an external reference dataset to train the classifier[43]. The widely used Percolator approach subsequently employed a semi-supervised learning approach that was trained directly on the dataset

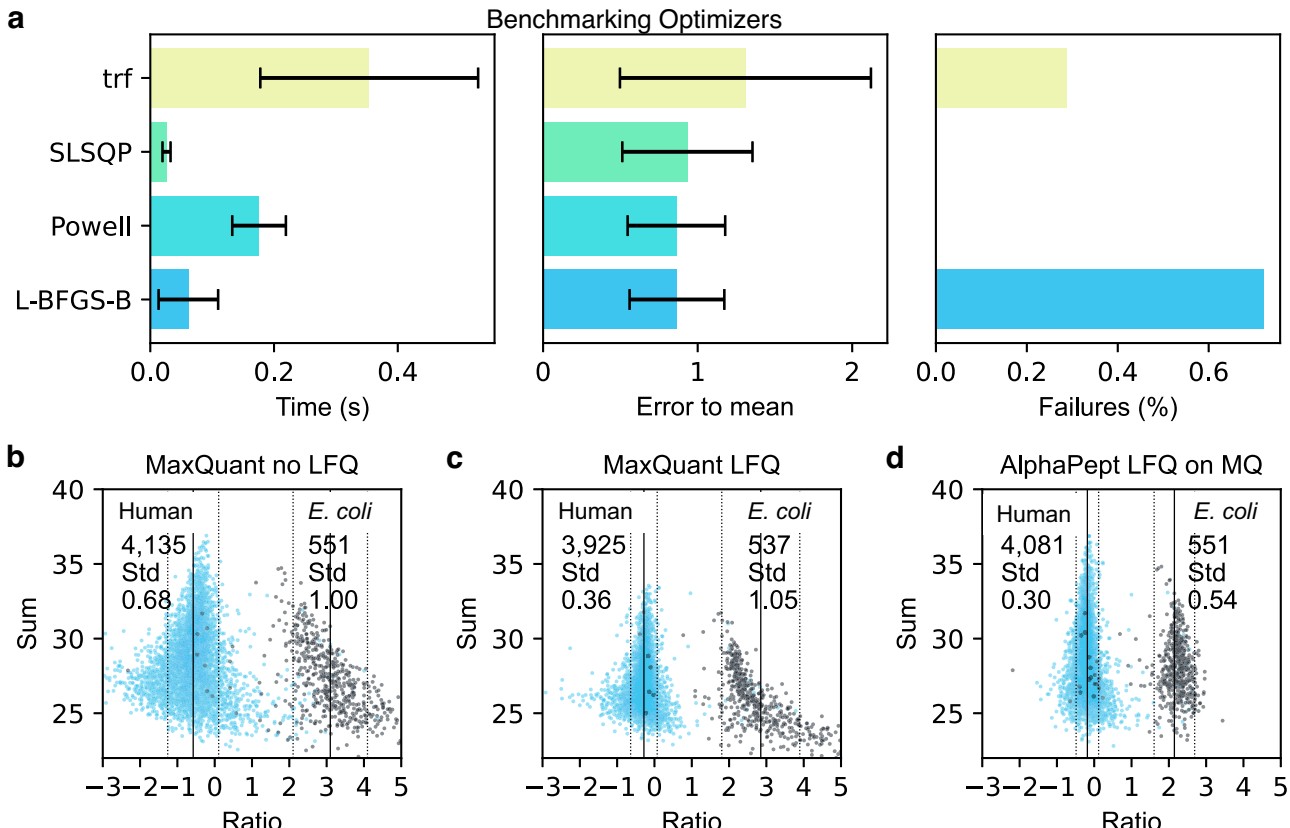

**Fig. 5 | Algorithm selection and performance of label-free quantification.**
**a** Timings of different, highly optimized solvers from the SciPy ecosystem to extract optimal protein intensity ratios in AlphaPept. Solvers showed drastic differences in speed, closeness to "ground truth", and proportion of successful optimizations on in silico test data. Each optimization was run repeatedly ($n = 200$), and error bars show one standard deviation. Based on these tests, AlphaPept employs a hybrid optimization strategy that uses L-BFGS-B and Powell for optimized performance, robustness, and speed. **b** Uncorrected MaxQuant intensities. **c** Intensity distributions after MaxQuant LFQ optimization. **d** Comparing the AlphaPept LFQ solver on MaxQuant output data demonstrates better separation in mixed-species datasets with smaller standard deviations and more protein groups retained.

itself[44]. This automatically adapts the ML model to the experimental data, and along with other MS analysis tools[45–49], we also employ semi-supervised learning for PSM scoring in AlphaPept.

The AlphaPept scoring module falls into five parts: (1) feature extraction for all candidate PSMs, (2) selection of a candidate subset, (3) training of a machine learning classifier, (4) scoring of all candidate PSMs, and (5) FDR estimation by a target-decoy approach (Fig. 4a). Most features for scoring the candidate PSMs are directly extracted from the search results, such as the number of b- and y-ion hits and the matched ion intensity fraction. Some additional features are subsequently determined, including the sequence length and the number of missed cleavages. After feature extraction, a subset of candidate PSMs is selected with an initial 1% FDR threshold based only on conventional scores, such as X!Tandem, Morpheus or our "generic score" (Fig. 4b). Together with an equal number of randomly selected decoys, this creates a balanced dataset for machine learning. This is split into training and test sets (20 vs. 80%) and provides the input of an ML classifier. We chose a standard scikit-learn random forest (RF) classifier as it performed similarly to XGBoost with fewer dependencies on other packages. We first identify optimal hyperparameters for the classifier with a grid search via five-fold cross-validation. The resulting best classifier optimally separates the target from decoy PSMs on the test set. Applying the trained classifier to the entire set of candidate PSMs yields discriminant scores that are used to estimate $q$ values based on the classical target-decoy competition approach.

The contribution of different features to the discriminant score for an exemplary tryptic HeLa sample is shown in Fig. 4c. Interestingly,

for our data, the number of matched y-ions alone outperforms the basic search engine score, and most of the top-ranking features are related to the number of matched ions and their intensity. The ML algorithm markedly improved the separation of targets vs. decoys, retrieving a larger number of PSMs at every $q$ value (Fig. 4d). ML-based scoring in AlphaPept improved identification rates by 14.4 at a 1% FDR at the PSMs level, in line with previous reports[44]. AlphaPept allows ready substitution of the underlying PSM score and machine learning algorithms. Furthermore, additional features to describe the PSMs are readily integrated, such as ion mobility or predicted fragment intensities. We envision that this kind of flexibility will enable continuous integration of improved workflows as well as novel ML techniques into AlphaPept. Our recently developed deep learning framework Alpha-PeptDeep allows seamless integration to learn properties from sequences based on AlphaPept search results[50].

Once a set of PSMs at a defined FDR is identified, protein groups are determined via the razor protein approach[51]. Here, peptides that could potentially map to multiple unique proteins are assigned to the protein group that already has the most peptide evidence. We determine protein-level $q$ values by selecting the best scoring precursor per protein, followed by FDR estimation by target-decoy competition similar to the peptide level[52–55]. Finally, we validated the scoring and FDR estimation in AlphaPept with the entrapment strategy mentioned above, by analyzing a HeLa sample with a mixed-species library, containing targets and decoys derived from both a human FASTA and a FASTA from Arabidopsis thaliana. This revealed that AlphaPept provides accurate q-value estimates, reporting approximately the same

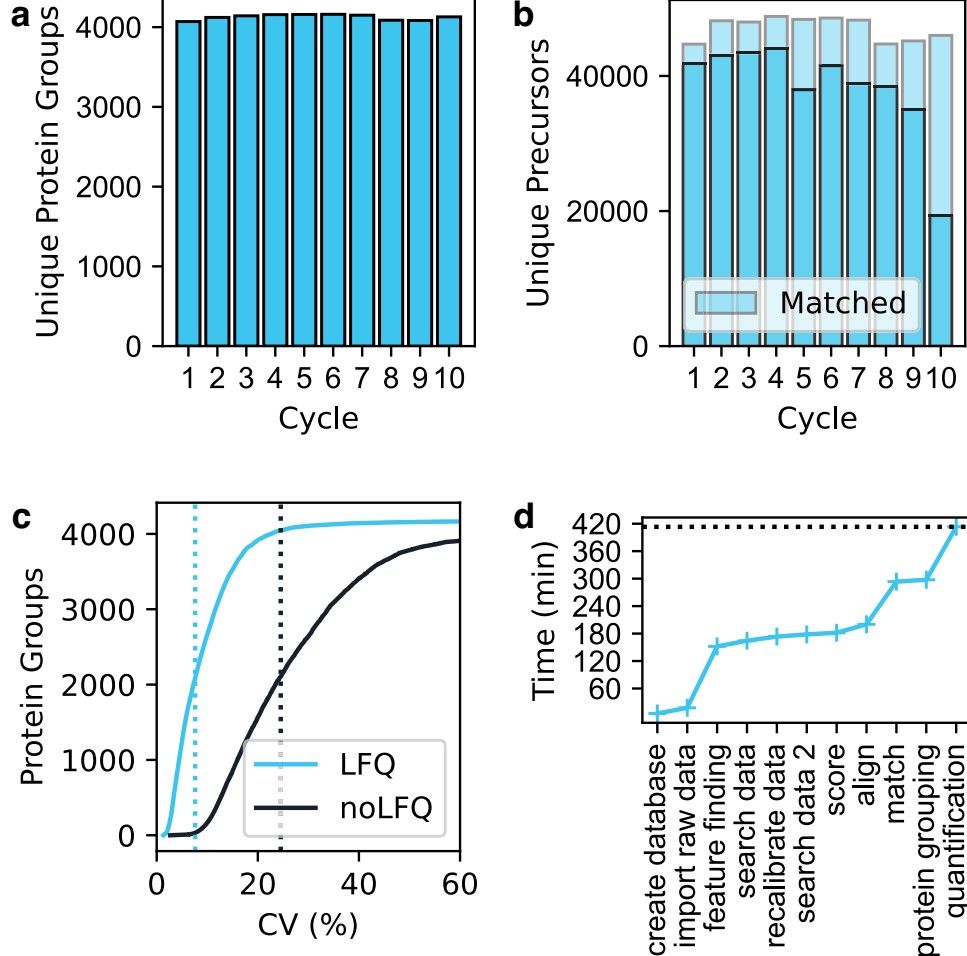

**Fig. 6 | Processing 200 HeLa proteomes with AlphaPept.** A total of 200 DDA HeLa cell proteomes – the ten-cycle long-term performance test from Kuster and coworkers (181 Gbyte from PXD015087)[59]– was analyzed by AlphaPept. **a** Identification performance at the protein group level. **b** Identification performance at the precursor level. **c** Quantification performance with or without MaxLFQ optimization. **d** Timing of the AlphaPept computational pipeline for the entire 200 HeLa proteome run. Search through scoring are highly optimized and contributes little to overall computation time.

number of *Arabidopsis thaliana* proteins as decoy proteins at 1% protein FDR (See also Supplementary Figs. 6, 7).

## Label-free quantification

The ultimate goal of a proteomics experiment is to derive functional insights or assess biomarkers from quantitative changes at the protein-level, to which peptide identifications are the only means to an end. Algorithmically this quantification step entails either the determination of isotope ratios in the same scans (for instance, SILAC, TMT, or EASI-tag ratios) or the somewhat more challenging problem of first integrating peaks and then deriving quantitative ratios across samples (label-free quantification), which we focus on here. We initially adapted the MaxLFQ approach for label-free quantitative proteomics data[56]. The first task is to determine normalization factors for each run as different LC-MS/MS (tandem mass spectrometry) runs need to be compared—potentially spaced over many months in which instrument performance may vary— and as total loading amounts likewise vary, for instance, due to pipetting errors. The basic assumption is that the majority of peptides are not differentially abundant between different samples. This allows deriving the run-specific normalization factors by minimizing the between-sample log peptide ratios[56] (Note that this assumption is not always valid and can be restricted to certain protein classes). In the second step, adjusted intensities are derived for each protein, such that protein intensities between different MS runs can be

compared. To this end, we derive the median peptide fold changes that maximize consistency with the peptide evidence.

The normalization, as well as protein intensity profile construction, are quadratic minimization problems of the normalization factors or the intensities, respectively. Such minimization problems can be solved in various ways, but one fundamental challenge is that these algorithms have a time complexity of $O(n^2)$, meaning that the computation time increases quadratically with the number of comparisons. One strategy to overcome this limitation is to only perform minimization on a subset of all possible pairs (termed "FastLFQ")[56]. Despite this, the computation time of the underlying solver will determine the overall runtime and account for the long run times on very large datasets. However, a variety of very efficient solvers that are based on different algorithms are contained in the Python SciPy package[57]. To test these approaches, we created an in silico test dataset with a known ground truth (see Quantification Notebook). Comparing different solvers using our benchmarking set repeatedly ($n = 200$) uncovered dramatic differences in precision, runtime, and success rate (Fig. 5a). Among the better-performing algorithms were the least-squares solvers that were previously used. The Broyden–Fletcher–Goldfarb–Shanno (L-BFGS-B), Sequential Least-Squares Programming (SLSQP), and Powell algorithms were particularly fast and robust solutions being more than 6x faster than the Trust Region Reflective algorithm (trf) from the default least-squares solver. More remarkably, they were able to optimize much

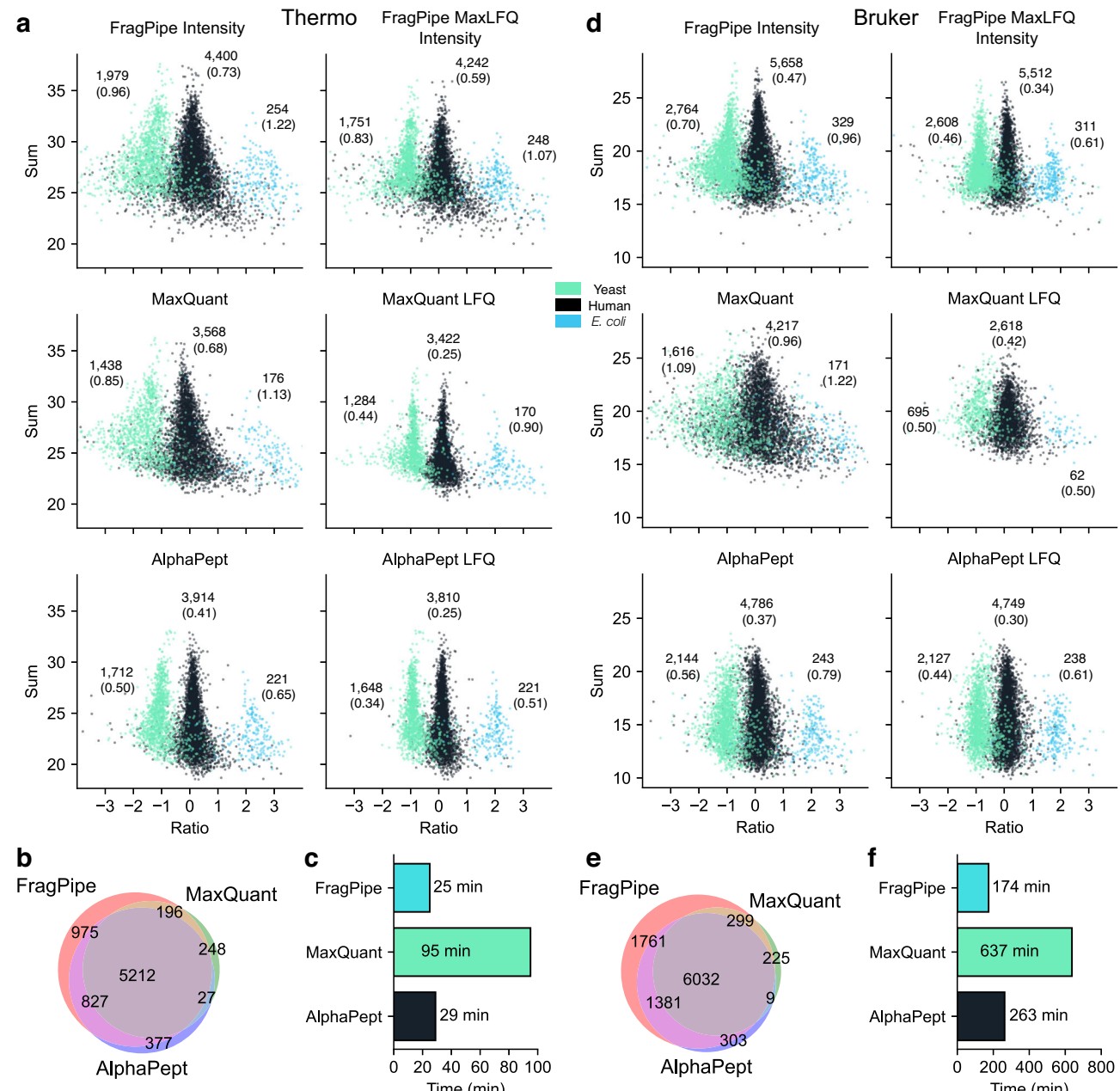

**Fig. 7 | Benchmarking AlphaPept on Thermo and Bruker mixed-species datasets. a** Mixed-Species analysis from PXD028735, with six Thermo files. The left columns show protein intensity ratios before LFQ optimization, Right columns show intensity ratios after LFQ optimization. Ratios and summed intensity are in logarithmic scale (log2). **b** Venn Diagram of all quantified proteins shows good agreement across all search engines. **c** Computation time. **d** Mixed-species analysis from PXD028735 with six Bruker files. **e** Venn Diagram of all quantified proteins shows good agreement across all search engines. **f** Computation time.

better to our known ground truth. Of all four tested optimizers, the mean absolute percentage error of trf was, on average twice as high. Being able to readily switch between different solvers provided by SciPy allows us to fall back on other solvers if the default solver fails, i.e., AlphaPept will switch from L-BFGS-B to Powell if the solution does not converge.

We compared our method to MaxLFQ in a quantitative two-species benchmarking dataset, in which *E. coli* proteins change their abundance by a factor of six between conditions, while human proteins do not change[58]. To specifically assess the benefits of the optimization strategy, we first tested the algorithm directly on the MaxQuant output (see companion Notebook for Fig. 5). Before LFQ optimization the distributions are already distinguishable (Fig. 5b).

After LFQ optimization, the separation of human and *E. coli* proteins clearly improved for both the MaxQuant LFQ optimization (Fig. 5c) as well as the AlphaPept LFQ optimization (Fig. 5d); furthermore, the standard deviation was smaller (0.36 vs. 0.30 for Human and 1.05 vs. 0.54 for *E. coli*) when applying the AlphaPept optimization algorithm, which also has fewer outlier quantifications, supporting the analysis of the in silico test set. Additionally, AlphaPept retained more proteins (4081 vs. 3925 from 4135). The total computation time for this optimization was 143 s.

**Match-between-runs (MBR) and dataset alignment**
We implemented functionality to transfer the identifications of MS1 features to unidentified MS1 features of other runs (match-between-

runs). MBR is a popular functionality in MaxQuant, but the code for the implementation of the underlying algorithm has not been made public. First, we align multiple datasets on top of each other by applying a global offset in retention time, mass, and—where applicable—ion mobility. To determine offsets for all runs, we first compare all possible pairs of runs and calculate the median offset from one dataset to another based on the precursors that were identified in both. As these offsets are linear combinations of each other, i.e., the offset from dataset A to dataset C should be the offset from dataset A to B and B to C; this becomes an overdetermined equation system, which we solve by a weighted linear regression model with the number of shared precursors as weights.

After dataset alignment, we group precursors of multiple runs, determine their expected properties as well as their probability density, and create a library of precursors. Next, we take the unidentified MS1 features from each run and extract the closest match from the library of precursors. Finally, as we know the probability density of each feature, we can calculate the Mahalanobis distance from each identification transfer and use this as a probability estimate to assess the likelihood that a match is correct. Further Information about the alignment and matching algorithm can be found in the Matching notebook.

## Processing 200 HeLa proteomes with AlphaPept

A prime goal of the AlphaPept effort is robustness and speed. To showcase the usability of AlphaPept for large-scale studies, we re-analyzed 200 HeLa proteomes from a recently published long-term performance test[59] on a processing workstation (Intel i9, 3.5 GHZ, 12 cores with 128 GB RAM). This dataset is available under ProteomeXchange accession number PXD015087. To confirm comparable identification performance in the initial analysis, which was done with MaxQuant, we evaluated the number of uniquely identified protein groups and PSMs per group. This yielded a mean of 4126 unique protein groups (Fig. 6a) and 47,082 unique precursors per experimentally defined group with matching (Fig. 6b). Note that the drop in identifications in Cycle 10 is also reported in the original study. For reference, the average number of protein groups identified with MaxQuant was (3849) but reported higher precursor numbers (55,585). Part of this difference is due to the stringency in matching; MaxQuant matches, on average, 12,067 precursors, whereas AlphaPept matches 8716. Next, we investigated protein-level quantification. The median coefficient of variation without our Python maxLFQ implementation was 24.5 and 7.6% after LFQ optimization. Investigation of each computational task revealed that a large part is spent on feature finding. During optimization, we noticed that disk speed is a highly contributing factor for processing times, highlighting that the algorithms are rather I/O bound than compute bound. Searching and scoring are highly optimized and contribute only a small fraction of the overall computing time. Operations across files, such as LFQ alignment and matching, again make up a larger part of computation time. In total, the search took 413 min, for an average of 2.07 min per file. For comparison purposes, we re-ran this dataset on various hardware configurations, such as multiple cloud instances and our SLURM cluster, resulting in total processing times from 280 min (cluster with preprocessed files, on average 1.4 min per file) to 482 min, further confirming the I/O bounds, see Supplementary Table 1.

## Benchmarking AlphaPept against other search engines

For an overview of how AlphaPept performs on an entire workflow, we benchmarked against the latest versions of popular search engines MaxQuant (2.1.4.0) and FragPipe (MSFragger 3.5 + Philosopher + IonQuant)[22,60–63] on a LFQ benchmark dataset with mixed species (*E. coli*, Human, and Yeast), that was previously published[64], available under ProteomeXchange accession number PXD028735. We tested conditions for Thermo Orbitrap and Bruker timsTOF with six files each. The dataset contains two conditions, with Humans in equal amounts

and Yeast and *E. coli* spiked in defined ratios and with three replicates for each condition. This allowed us to test the entire workflow and gave robust insight into the overall quantification and identification performance of each search engine. We used the default AlphaPept settings without MBR and an LFQ minimum ratio of 1. We note that matching settings between search engines is inherently difficult, and each search engine could perform relatively better or worse with other settings; therefore we chose to change the default settings as little as possible.

For the Thermo test set, FragPipe found the most quantifiable proteins, whereas MaxQuant and AlphaPept showed more defined distributions (smaller standard deviation for both species) and fewer outlier proteins (Fig. 7a). We denote quantifiable proteins as proteins detected in enough repeats so that a signal can be calculated with the LFQ algorithm. Supplementary Fig. 4 shows the accuracy in comparison to the expected ratio over the number of identified proteins, where the performance of AlphaPept is between MaxQuant (worse) and MSFragger (better). A Venn diagram of all quantified proteins across all runs showed excellent agreement between all search engines (Fig. 7b). Timingwise, FragPipe, and AlphaPept showed similar performance, with FragPipe being slightly faster and both approximately three times as fast as MaxQuant (Fig. 7c). For the Bruker test set, again FragPipe found the most quantifiable proteins. AlphaPept and FragPipe showed similar distributions, and MaxQuant somewhat worse ones (Fig. 7d). AlphaPept was 34% slower than FragPipe but took less than half of the time of MaxQuant (Fig. 7f). Supplementary Fig. 5 shows the deviation to the expected ratio over the number of identified proteins, where the performance of AlphaPept is between MaxQuant and MSFragger. A Venn diagram of all quantified proteins across all runs showed excellent agreement between all search engines (Fig. 7e). This indicates that AlphaPept can produce comparable search results when compared to other frequently used state-of-the-art software under standard conditions.

## Continuous validation of standard datasets

Our current continuous integration pipeline uses a range of datasets typical for DDA (data-dependent acquisition) MS workflows. These include standard single-shot runs, such as HeLa quality control (QC) runs, as well as recently published studies. For every addition to the main branch of the code base, AlphaPept reanalyzes these files fully automatically, allowing extensive systems checks. Additionally, these checks can be manually triggered at any time and therefore enable swift validation of proposed code changes prior to submitting pull requests. This makes comparing studies that were analyzed with different software versions much more transparent. To further increase this idea of transparent performance tracking, we automatically upload summary statistics, such as runtime, number of proteins, and number of features for each run to a database and visualize these metrics in a dashboard (Extended methods). Table 1 shows example tracking metrics from the database.

## AlphaPept user interface and server

A central element for any software tool is ease of use for the end user. In the most basic setup, this is determined by the accessibility of the GUI. Following recent trends, we decided on server-based technology for AlphaPept. In a basic setup, the web interface is called by connecting to a local server instance on the user's laptop or local workstation (Fig. 8a) via a browser. For more demanding pipelines, AlphaPept can be run on a powerful processing PC that is accessed from multiple other devices. This makes access to AlphaPept independent and it can even be used from mobile devices.

Adding server functionality typically comes at the cost of maintaining a dedicated API and infrastructure. For AlphaPept we make use of a very recent but already very popular Python package called streamlit (www.streamlit.com), which was developed to facilitate the sharing of machine learning models. By only adding one additional

**Table 1 | Example performance tracking metrics for different AlphaPept versions extracted from the database**

| Version | Test file | Processing time (min) | Number of features | Number of peptides |
|---------|-----------|----------------------|--------------------|--------------------|
| 0.2.8 | HeLa Orbitrap | 19 | 218,792 | 41,777 |
| 0.2.8 | HeLa timsTOF | 102 | 231,545 | 54,058 |
| 0.2.9 | HeLa Orbitrap | 19 | 218,780 | 41,939 |
| 0.2.9 | HeLa timsTOF | 113 | 231,545 | 66,776 |
| 0.2.10 | HeLa Orbitrap | 19 | 218,779 | 41,949 |
| … | … | … | … | … |
| 0.3.25 | HeLa timsTOF | 105 | 664,992 | 76,217 |
| 0.3.26 | HeLa Orbitrap | 18 | 260,709 | 53,522 |
| 0.3.26 | HeLa timsTOF | 88 | 664,992 | 77,464 |
| 0.3.27 | HeLa Orbitrap | 21 | 260,622 | 54,283 |
| 0.3.27 | HeLa timsTOF | 89 | 664,992 | 77,162 |

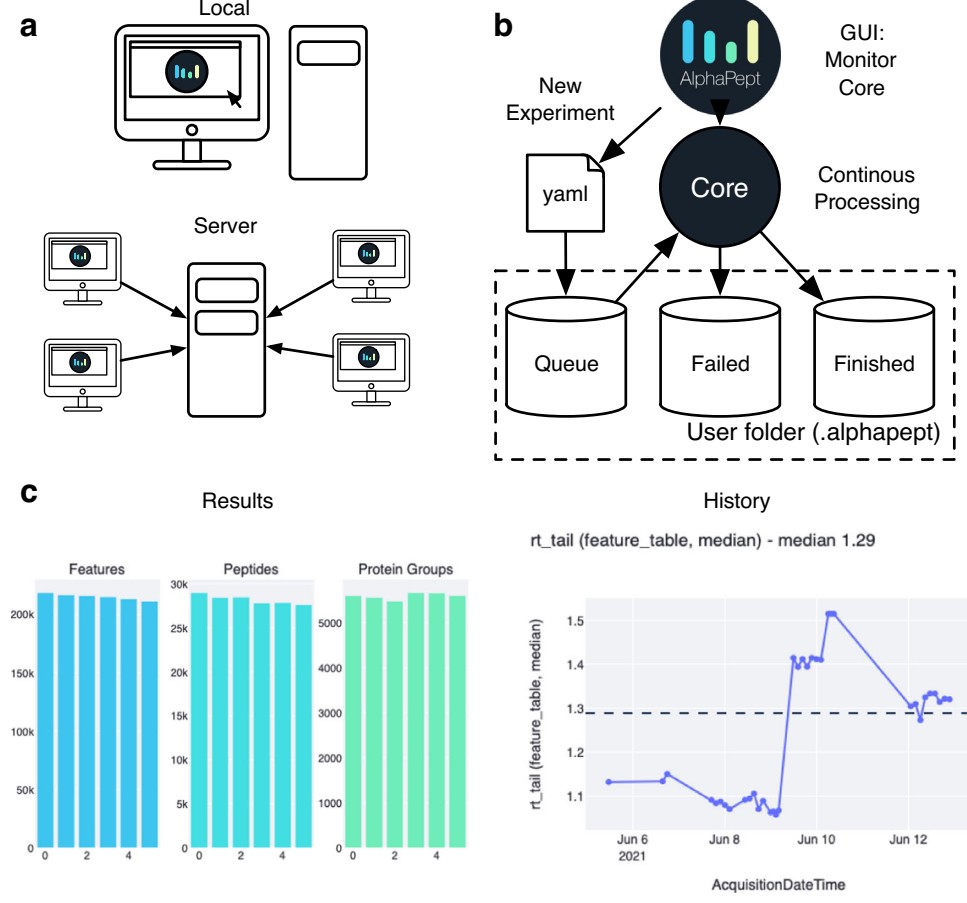

**Fig. 8 | Alphapept user interface, workflow management, deploying, and integrating. a** The AlphaPept GUI is based on a server architecture that can be installed on a workstation and used locally. Additionally, it can be installed on a server and accessed remotely from multiple workstations in the network. **b** AlphePept processing pipeline. The AlphaPept GUI creates three folders for its processing system. New experiments are defined within the interface and saved as YAML files in the Queue folder with automatically triggered processing. **c** Example plots from the History and Results Tab in AlphaPept: Overview of the number of features, peptides, and protein groups per injected sample (left panel). Graphing retention time tailing as a function of acquisition date, as an illustration of using AlphaPept for quality assurance.

Python package, we have access to a powerful and responsive server infrastructure. Here, the web interface serves merely as an input wrapper to gather the required settings and display results and starts the AlphaPept processing in the background.

## AlphaPept workflow management system
Importantly, the server-based user interface extends the processing functionality of AlphaPept from only processing individual experiments to a continuous processing and monitoring framework. The core processing function of AlphaPept accepts a dictionary-type document to process an experiment, with defined parameters per setting. To store these settings, we chose YAML, a standard human-readable data-serialization language, resulting in files of only a few kilobytes in size. This ensures that they can be modified programmatically and easily checked with common editors.

The settings structure is used by the AlphaPept GUI to build a folder-based workflow management system. It creates three folders in the user folder ("Queue", "Failed", and "Finished") and monitors them for new data. When defining a new experiment within the GUI, a settings YAML file is created in the Queue folder, and the core function starts processing. This allows for defining multiple experiments, which will then be processed one after another. YAML files of processed runs will be moved to the "Finished" or "Failed" folder (Fig. 8b).

We chose this folder-based processing queue as this allows manual inspection of the processing queue by simply checking the files in the folders. Furthermore, computational alterations of the processing queue are straightforward by writing custom scripts that copy settings files generated elsewhere to the queue folder. AlphaPept has a file watcher module that can monitor folders for new raw files and automatically add them to the processing queue immediately after the acquisition is finished. Its modular structure can easily be extended with custom code for integration into larger processing environments with database-based queuing systems. The interface notebook demonstrates the calls to the wrapper function and allows customization of the pipeline.

### Visualization of results and continuous processing

For visualization of tabular or summary statistics results, our streamlit application utilizes the "Finished" folder structure, where it stores readily accessible summary information of previously processed files (Fig. 8c). AlphaPept has a History tab that compiles these previous results to show performance over time or across analyzed MS runs (Fig. 8d). Here, the user can choose to plot various summary statistics such as identified proteins or peptides as well as chromatographic information such as peak width or peak tailing. As a particular use case, this provides a standard interface which allows instant QC run evaluation in combination with the file watcher.

To inspect an individual experiment, AlphaPept's browser interface can also plot identification and quantification summary information. Furthermore, basic data analysis functions such as volcano or scatter plots and Principal Component Analysis (PCA) are provided. This is based on streamlit and scikit-learn functionality and can therefore be readily extended. AlphaPept exports the analysis results (quantified proteins and peptides) in tabular format to the specified results path so that it can be readily used for other downstream processing tools such as Perseus[16] or our recently introduced CKG[15].

### AlphaPept deployment and integration

The utility of a computational tool critically depends on how well it can be integrated into existing workflows. To enable maximum flexibility and to address all major use cases, AlphaPept offers multiple ways to install and integrate it.

First, we provide a one-click installer solution that is packaged for a standard Windows system obviating additional installation routines. It provides a straightforward interface to the web-based GUI. We chose Windows for the one-click solution as it is the base OS for the vendor-provided acquisition and analysis software and most users. The one-click installation also has a command-line interface (CLI) for integration into data pipelines.

Next, AlphaPept can be used as a module in the same way as other Python packages. This requires setting up a Python environment to run the tool, which also contains all the functionality of the previously described CLI and GUI. Compared to the Windows one-click installer, the Python module extends the compatibility to other operating systems. While Python code is, in principle, cross-platform, some third-party packages can be platform-bound, such as the Bruker feature finder or DLLs required to read proprietary file types. The modular nature of the AlphaPept file system allows us to preprocess files and continue the analysis on a different system (e.g., feature finding and file conversion on a Windows acquisition PC and processing on a Mac system).

Finally, the Python module makes the individual functions available to any Python program. This is particularly useful to integrate only parts of a workflow in a script or to optimize an individual workflow step. Besides the nbdev notebooks that contain the AlphaPept core code, we provide several sandboxing Jupyter Notebooks that show how individual workflow steps can be called and modified. In this way, AlphaPept allows the creation of completely customized workflows.

For a typical proteomics laboratory running DDA experiments, we envision AlphaPept running in continuous mode, to automatically process all new files. This allows continuous feedback about experiments while drastically speeding up computation when subsequently combining multiple processed files into experiments and these experiments into an overall study. This is because the computational steps that do not change (e.g., raw conversion, database generation, or feature finding) can be reused.

Being able to import AlphaPept as a Python package also lowers the entry barrier of proteomics analysis workflows for individual researchers and laboratories with little computational infrastructure, as it makes it compatible with platforms like Google Colab, a free cloud-based infrastructure built on top of Jupyter notebooks with GPUs. This allows processing, subject to resource availability from Google, without having to set up software on specialized hardware and allows direct modification of the underlying algorithms. We provide an explanatory notebook for running a workflow on Google Colab, including a 120 min HeLa example file that has been converted on the Windows acquisition computer. This also highlights how the modular HDF5 file format allows us to move the MS data between operating systems. As another showcase of AlphaPept's portability, AlphaPept was integrated to the reproducible data science platform Renku by the community[65].

## Discussion

Here we have introduced AlphaPept, a computational proteomics framework where the relevant algorithms are written in Python itself, rather than Python being used only as a scripting layer on top of compiled code. This architectural choice allows the user to inspect and even modify the code and enables seamless integration with the tools of the increasingly powerful and popular Python scientific ecosystem. The major drawback of such an approach would have been the slow execution speed of pure Python; however, extensive use of the Numba just-in-time compiler—on multiple CPUs or a GPU—makes AlphaPept exceptionally fast, as we have shown in this manuscript. Together with the use of recently developed browser-based deployment, AlphaPept covers the full range of potential users, from novice users to systems administrators wishing to build large cloud pipelines.

A related and important design objective of AlphaPept was to enable a diverse user community and invite community participation in its further development. To ensure quality, reproducibility, and stability, we implemented a large suite of mechanisms from the unit through end-to-end tests via automatic deployment tools. This, in turn, allows us to streamline the integration of community contributions after rigorous assessment. Furthermore, GitHub provides state-of-the-art tools and mechanisms to allow the effective collaboration of diverse and dispersed developer communities.

Currently, AlphaPept provides functionality for DDA proteomics, but we are in the process of enabling analysis of DIA data, ultra-fast access to and visualization of ion mobility data (AlphaTims[26], https://github.com/MannLabs/alphatims), deep learning for predicted peptide properties (AlphaPeptDeep[50], https://github.com/MannLabs/alphapeptdeep), Visualization of search engine results (AlphaViz, https://github.com/MannLabs/alphaviz), visual annotation of results (AlphaMap[66,67], https://github.com/MannLabs/alphamap), statistical downstream analysis (AlphaPept

Stats[68]) and improved quantification, all made possible by its modular design. While AlphaPept may not be best-in-class across all benchmarks, it is on par with widely used proteomics tools but has the benefit of being completely open and permissively licensed. This makes it an excellent educational resource for future algorithmic upgrades which can be modularly integrated.

Recent acquisition schemes allow precisely assigning precursor to fragmentation data, even from DIA acquisition schemes. This will allow creating not only "pseudo" but also "pure spectra" which can be searched with conventional DDA search engines (DIA Umpire)[69-71]. As a result, we believe that DDA search engines will soon be a routine part of DIA workflows as well. Notably, the speed of AlphaPept will be crucial for these "DDA from DIA" approaches because they produce many-fold larger datasets to search compared to DDA.

One of the large goals of AlphaPept is to "democratize" access to computational proteomics. To this end, besides implementation in Python, we adopted the 'literate programming' paradigm which integrates documentation and code. We adopted the nbdev package, providing both beginner and expert computational proteomics researchers with an easy and interactive "on ramp". In our case, this takes the form of currently 12 Jupyter notebooks dealing with all the major sub-tasks of the entire computational pipeline, from database creation, raw data import all the way to the final report of the results. We imagine that students and researchers with innovative algorithmic ideas can use this paradigm to add their functionality in a transparent and efficient manner, without having to re-create the entire pipeline. This could especially enable increasingly powerful machine learning and deep learning technologies to be integrated into computational proteomics[7,8,72,73].

## Methods

*MongoDB Dashboard* The continuous integration pipeline has the action "Performance test pyinstaller". This action freezes the current Python environment into an executable and runs the test files. The results of these tests are uploaded to a noSQL database (MongoDB) for the tested version number. Key performance metrics are visualized in charts here:

https://charts.mongodb.com/charts-alphapept-itfxv/public/dashboards/5f671dcf-bcd6-4d90-8494-8c7f724b727b.

### timsTOF and Orbitrap HeLa samples

The test files comprise representative single-run analyses of complex proteome samples. Human HeLa cancer cells were lysed in reduction and alkylation buffer with chloroacetamide as previously described[74], and proteins were enzymatically digested with LysC and trypsin. The resulting peptides were de-salted and purified on styrene-divinylbenzene reversed-phase sulfonate (SDB-RPS) StageTips before injection into an EASY nLC 1200 nanoflow chromatography system (Thermo Scientific). The samples were loaded on a 50 cm × 75 μm column packed in-house with 1.9 μm $C_{18}$ beads and fitted with a laser-pulled emitter tip. Separation was performed for 120 min with a binary gradient at a flow rate of 300 nL/min. The LC system was coupled online to either a quadrupole Orbitrap (Thermo Scientific Orbitrap Exploris 480) or a trapped ion mobility−quadrupole time-of-flight (Bruker timsTOF Pro 2) mass spectrometer. Data were acquired with standard data-dependent top15 (Orbitrap) and PASEF (Parallel Accumulation−Serial Fragmentation) methods (timsTOF), respectively.

### timsTOF and Orbitrap iRT samples

Eleven iRT peptides (https://biognosys.com/product/irt-kit/) were separated via a 5.6 min Evosep gradient (200 "samples per day"), yielding test data with low complexity, that facilitated quick testing of computational functionality. An Evosep One liquid chromatography system (Evosep) was coupled online with a trapped ion mobility spectrometry (TIMS) quadrupole time-of-flight (TOF) mass spectrometer (timsTOF pro, Bruker Daltonics). iRT standards (Biognosys) were loaded onto Evotips according to the manufacturers' instructions and separated with a 4 cm × 150 μm reverse-phase column with 3 μm $C_{18}$-beads (Pepsep). The analytical column was connected with a zero-dead volume emitter (10 μm) placed in a nano-electrospray ion source (CaptiveSpray source, Bruker Daltonics). Mobil phase A contained 0.1 vol% formic acid and water, and mobile phase B of 0.1 vol % formic acid and acetonitrile. The sample was acquired with the dda-PASEF acquisition mode. Each Top-N acquisition mode contained four PASEF MS/MS scans, and the accumulation and ramp time were both 100 ms. Only multiply charged precursors over the intensity threshold of 2500 arbitrary units (a.u.) and within a m/z-range of 100 – 1700 were subjected to fragmentation. Peptides that reached the target intensity of 20,000 a.u. were excluded for 0.4 min. The quadrupole isolation width was set to 2 Th below *m/z* of 700 and 3 Th above a *m/z* value of 700. The ion mobility (IM) range was configured to 0.6–1.51 Vs cm$^{-2}$ and calibrated with three Agilent ESI-L TuneMix Ions (*m/z*, IM: 622.02, 0.98 Vs cm$^{-2}$; 922.01, 1.19 Vs cm$^{-2}$; 1221.99, 1.38 Vs cm$^{-2}$). The collision energy was decreased as a function of the ion mobility, starting at 1.6 Vs cm$^{-2}$ with 59 eV and ending at 0.6 Vs cm$^{-2}$ with 20 eV.

The results in this manuscript were obtained with AlphaPept version 0.5.0 if not otherwise indicated. Runtime is for CPU unless indicated otherwise.

During the revisions of this work, the authors used ChatGPT-4 by OpenAI in order to improve the readability and conciseness of the manuscript. After using this tool/service, the authors reviewed and edited the content as needed and take full responsibility for the content of the publication.

### Reporting summary

Further information on research design is available in the Nature Portfolio Reporting Summary linked to this article.

## Data availability

Data for benchmarking was taken from public repositories PXD015087 (High-flow chromatography for reproducible and high-throughput quantification of proteomes) and PXD028735 (A comprehensive LFQ benchmark dataset to validate data analysis pipelines on modern-day acquisition strategies in proteomics). Test files for continuous testing were uploaded to the Max-Planck data share, accessible via links present on test_ci.py in the GitHub repository at https://github.com/MannLabs/alphapept/blob/master/test_ci.py. Additional benchmarking results and data for SI Table 1 is available on Zenodo as https://doi.org/10.5281/zenodo.10223453.

## Code availability

AlphaPept is fully open-source and is freely available under an Apache license at https://github.com/MannLabs/alphapept. All notebooks are part of the repository in the "nbs" folder. The documentation created based on the notebooks is available here: https://mannlabs.github.io/alphapept/. Additional information about code not covered in the Notebooks presented here can be found in the Documentation (https://mannlabs.github.io/alphapept/additional_code.html).

A cloud-hosted Notebook with an example data file is provided at the free Google Colab site:

https://colab.research.google.com/drive/163LTlyzBCDgyCkSJiikbmsnny_EiQ7SG?usp=sharing.

All code to reproduce the figures in the manuscript is available on the GitHub repository.

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

## Acknowledgements

We thank Sven Brehmer, Wiebke Timm, Konstantin Schwarze, and Sebastian Wehner from Bruker Daltonik for providing support with the feature finder for Bruker data. Further, we thank Andreas Brunner, Igor Paron, Patricia Skowronek, and Mario Oroshi for providing sample files and descriptions and feedback on the QC pipeline. Xie-Xuan Zhou contributed to discussions and testing. We are grateful for the feedback, testing, and support from our colleagues and contributors from the open-source community. MTS is supported financially by the Novo Nordisk Foundation (Grant agreement NNF14CC0001). This study was supported by The Max-Planck Society for Advancement of Science and by the Bavarian State Ministry of Health and Care through the research project DigiMed Bayern (www.digimed-bayern.de).

## Author contributions

M.M. and M.T.S. conceived the core idea of the AlphaPept framework and M.M. wrote the first iteration of the search algorithm. F.M. contributed guidance for integrating Bruker support. M.T.S. wrote the Thermo feature finder, quantification and downstream processing modules, code structure, and user interface. E.V. contributed file importing functionality. I.B. extended the scoring functionality with M.L. and F.D.R. control. S.W. added HDF file handling, revised the general code structure, and added performance functions. W.F.Z. and C.A. contributed and improved quantification. J.S. critically reviewed testing and documentation. R.I. and M.G. contributed to GPU support and code acceleration. All authors contributed ideas, performed testing, and wrote the manuscript.

## Funding

## Competing interests

The authors declare the following competing interests: R.I. and M.G. are employees of Nvidia Corporation. M.T.S. was an employee of OmicEra Diagnostics GmbH. M.M. is an indirect investor in Evosep. The remaining authors declare no competing interests.
