## [Peer Review File · Nature Communications]

REVIEWER COMMENTS

Reviewer #1 (Remarks to the Author):

In this manuscript, the authors describe an open-source bottom-up data proteomics data analysis tool, AlphaPept. The software is written in Python, and heavily relies on Python packages, such as Numba, scikit-learn, nbdev, pandas, CuPy, etc. The software was well developed and tested. The authors performed several experiments to demonstrate the performance. The manuscript was also well written and organized. However, the authors failed to show the advantage of AlphaPept compared to the other state-of-the-art tools. It was not significantly faster than other tools (rather, it was slower in analyzing the Orbitrap data compared to MSFragger, Figure 8). It also quantified fewer proteins compared to MSFragger (Figure 8) although the authors claimed that the quantification quality is higher.

In terms of scientific novelty, AlphaPept is heavily based on the existing methods. The score function, X!Tandem, used in database searching has been proposed since 2003. AlphaPept uses the existing semi-supervised learning approach, which was proposed by Käll et al in 2007, to rescore PSMs. The label-free quantification was re-implemented based on MaxQuant (published in 2008). Furthermore, the speed-wise advantage over MaxQuant is also not due to the authors' novel contribution, rather, it is due to the usage of a popular python package, Numba (initially released in 2012).

In the introduction section, unfortunately, the authors failed to describe the challenges in the bottom-up proteomics data analysis. As a result, I could not clearly get what scientific problems they wanted to address by developing this new software.

Besides the above issues, there are also several major and minor issues (see the following lists). I also have a small suggestion at the end, and hopefully, the authors could add the phosphoproteome and HLA data analysis if they have time.

Major issues:

1. In the "peptide spectrum matching" section, the authors described that AlphaPept uses two stages of scoring: the first scoring only counts the matched fragments. Then, top-N peptides (N=10 by default) are used to calculate X!Tandem score. The reason for using fragment counting is for fast speed. And the reason for using X!Tandem score is to pick the most correct peptides. Since the first scoring uses a not-so-good score, there will be false negatives that are not in the top 10 candidates. To demonstrate that the two stages approach does not sacrifice the sensitivity too much, the authors should show the Venn

diagram of identified PSMs between the two-stage approach and the one-stage approach (all peptides are scored using X!Tandem score).

2. In the runtime comparison of Figure 8c and 8f, did AlphaPept use GPU? If not, the authors should add the runtime with GPU since they emphasized the advantage in multiple places. If yes, the authors should add the runtime without GPU since it is the case for most users.

3. The authors should upload the raw result files, parameter files, and log files of AlphaPept, MaxQuant, MSFragger, IonQuant, and FragPipe to Zenodo or figshare as required by the publication policy. It also makes the readers easier to evaluate and verify the experiment results.

4. In Figure 8, I believe MSFragger can only perform database searching. I guess the FDR filtering and label-free quantification were performed by Philosopher and IonQuant in FragPipe. The authors should clarify the tools they used in the benchmarking. And also cite those tools' papers:

a. Kong, A. T., Leprevost, F. V., Avtonomov, D. M., Mellacheruvu, D., & Nesvizhskii, A. I. (2017). MSFragger: ultrafast and comprehensive peptide identification in mass spectrometry-based proteomics. *Nature methods*, 14(5), 513-520.

b. Teo, G. C., Polasky, D. A., Yu, F., & Nesvizhskii, A. I. (2020). Fast deisotoping algorithm and its implementation in the MSFragger search engine. *Journal of proteome research*, 20(1), 498-505.

c. da Veiga Leprevost, F., Haynes, S. E., Avtonomov, D. M., Chang, H. Y., Shanmugam, A. K., Mellacheruvu, D., ... & Nesvizhskii, A. I. (2020). Philosopher: a versatile toolkit for shotgun proteomics data analysis. *Nature methods*, 17(9), 869-870.

d. Yu, F., Haynes, S. E., Teo, G. C., Avtonomov, D. M., Polasky, D. A., & Nesvizhskii, A. I. (2020). Fast quantitative analysis of timsTOF PASEF data with MSFragger and IonQuant. *Molecular & Cellular Proteomics*, 19(9), 1575-1585.

e. Yu, F., Haynes, S. E., & Nesvizhskii, A. I. (2021). IonQuant enables accurate and sensitive label-free quantification with FDR-controlled match-between-runs. *Molecular & Cellular Proteomics*, 20.

5. In Figure 8, the authors should label "MSFragger+Philosopher+IonQuant" or "FragPipe" if they used those tools in the benchmarking. Also, correct the description in the main context.

Minor issues:

1. In Figure 3a, what does the "Numba, threaded" means? The authors should explain the meaning in the figure caption.

2. In line 283, there is no Fig 3d in the manuscript. Did the authors miss one figure?

3. In line 341, the authors should cite two earlier publications:

a. Granholm, V., Navarro, J. F., Noble, W. S., & Käll, L. (2013). Determining the calibration of confidence estimation procedures for unique peptides in shotgun proteomics. *Journal of Proteomics*, 80, 123-131.

b. Feng, X. D., Li, L. W., Zhang, J. H., Zhu, Y. P., Chang, C., Shu, K. X., & Ma, J. (2017). Using the entrapment sequence method as a standard to evaluate key steps of proteomics data analysis process. *BMC Genomics*, 18(2), 1-9.

4. In line 355, Fig 4A is not about the target-decoy approach. I guess it should refer to Fig. 5A.
5. In line 360, it should refer to Fig. 5B, not Fig. 4B.
6. In line 370, it should refer to Figure 5c, not Figure 4c.
7. In line 373, it should refer to Fig. 5d, not 4d.
8. In line 577, it should refer to Fig. 9a, not 8a.
9. In line 602, it should refer to Fig. 9b, not 8b.
10. In line 615, it should refer to Fig. 9c, not 8c.
11. In line 617, it should refer to Fig. 9d, not 8d. The authors also forgot to label “9d” in the figure.
12. In line 671, it should refer to Table 2, not Table 1.

Suggestion:

1. It would be great if the authors could demonstrate AlphaPept’s performance in analyzing phosphoproteome and HLA datasets.

Reviewer #2 (Remarks to the Author):

Authors Strauss et al in their manuscript “AlphaPept, a modern and open framework for MS-based proteomics” revisit fundamental proteomics data processing with specific considerations for big data, cloud-based environments. In the provided demonstrations comparing AlphaPept to existing tools, AlphaPept does not typically outperform other search tools (particularly seems to underperform compared to the popular tool MSFragger), however AlphaPept is open source and written in Python, which is admittedly a major draw for some researchers in the community. After testing the software myself, I do think that AlphaPept will struggle to gain more users compared to MSFragger unless AlphaPept introduces some major draw for the average proteomicist. There are many other open source MS-based proteomics tools and frameworks – crux, comet, MSGF+, OpenMS – but I do not think it necessary to test every other software, and the two softwares tested here are probably the most popular used in the community for DDA.

Overall, the main novel aspect and noteworthy result of this work seems to be the language (Python) that it is written in.

MINOR CONCERNS:

- In the timings table, it's odd to see that cloud-based searching takes so much longer than a local machine. The resources on the cloud-based search are more on-par with the laptop resources, which seems like a strange choice for demonstrating cloud-based deployment. I would like to see how AlphaPept works in a truly powerful cloud environment, as I imagine deployment on AWS or other cloud services would be a major use case for the type of users drawn to a tool like this. Could the authors repeat the cloud-based runs with better AWS resources, as a major advantage of their software is purportedly that can be deployed in a scalable cloud environment?

- Colab notebook booted up fine for me, but was running slow and eventually crashed after using all available RAM. Could there be some comment added to the notebook describing how to allocate more RAM for this so that the demonstration completes?

- Installation of AlphaPept with the Windows installer worked fine for me, but I found the tool a bit clunky, at least in the example vignette provided. The navigation is a bit clunky – for example, selecting which files to be run from a folder if there's multiple files in the folder. I suppose that this is an argument for proper directory/subdirectory organization, but I can easily imagine situations where files from across directories are run and it would be a pain to copy-paste the files into multiple directories. It would be nice to provide another vignette example that utilizes the other Results plots (Volcano plot, Correlation heatmap, PCA). Could the authors include a second vignette example dataset, perhaps something more typical of a DDA search, to demonstrate those plotting functions?

Reviewer #3 (Remarks to the Author):

Strauss et al. describe a processing pipeline for mass spectrometry-based proteomics data that they named AlphaPept. The pipeline is benchmarked to run fast, despite being mostly developed in python code, a feature that the authors attribute to the numba python compiler. The pipeline is modularly built around a framework nbdev (<https://nbdev.fast.ai/>) which separates code and documentation from jupyter notebooks (Normally notebooks are great for visualization but tedious to program in). The nbdev allows for easy for inspection of the author's code but also allows the user to further add code in a straightforward manner.

The manuscript reads mostly as a walkthrough of the features of AlphaPept, and there is no actual novelty to the algorithmics. Despite its name, AlphaPept does not contain any deep learning. That said I

am very pleased with the efforts. The contribution of the manuscript lies in that it describes the engineering behind a series of nice feature implementations and optimizations. Also, neither of the compared pipelines, MSFRagger and MaxQuant are open source or transparent to their inner workings. Hence this manuscript is a step forward compared to such pipelines. I am, however, aware of the current rise of a series of different pipeline implementations inside of the NextFlow environment, which to me seem as transparent and are likely easier to install and execute than AlphaPept. The rise of such pipelines should be mentioned in the introduction.

When investigating the running times in Table 2, I am missing a comparison point. As an academic researcher, my alternative to process data is not to pay for an AWS cloud. Instead, I use my university cluster which provides a SLURM queue, and I containerize my software using e.g. Singularity to assure smooth execution over 100s of cores in parallel. Please include a comparison to processing in such environments.

The benchmark against MSFRagger and MaxQuant is apt, however, it is hard to get the conclusions from Figure 8. The ratio/sum scatter plots are hard to compare with the naked eye. Add a direct comparison of the number of reported differentially abundant peptides as a function of FDR for the three methods. Also, you have to spell out what is meant by “quantified” peptides and proteins in the Venn diagrams of Figures 8b&8e, e.g. are they above a FDR threshold? A FC Treshold?

REVIEWER COMMENTS

AlphaPept, a modern and open framework for MS-based proteomics

Reviewer #1 (Remarks to the Author):

In this manuscript, the authors describe an open-source bottom-up data proteomics data analysis tool, AlphaPept. The software is written in Python, and heavily relies on Python packages, such as Numba, scikit-learn, nbdev, pandas, CuPy, etc. The software was well developed and tested. The authors performed several experiments to demonstrate the performance. The manuscript was also well written and organized. However, the authors failed to show the advantage of AlphaPept compared to the other state-of-the-art tools. It was not significantly faster than other tools (rather, it was slower in analyzing the Orbitrap data compared to MSFragger, Figure 8). It also quantified fewer proteins compared to MSFragger (Figure 8) although the authors claimed that the quantification quality is higher.

We thank the author for the positive comments about the manuscript and software quality. The advantage of a proteomics package can indeed be seen in identification, speed and quantification, where we see AlphaPept in line with widely used tools, although maybe not best in class. A major advantage of AlphaPept as it is entirely open and freely usable without any restrictions, which is of great interest to parts of the community, as also noted by Reviewer 2 and 3.

For the revision, we have updated parts of the code and provided a more extensive benchmark set. In these measurements, FragPipe-MSFragger is still the fastest in class. We speculate that this is due to the conceptually different search algorithm being used, which has an edge in identification, especially for open searches. However, we note that AlphaPept is modular and the algorithm can in principle be replaced, for instance by the MSFragger algorithm. The current search algorithm is a rather conventional fragment-counting algorithm. Implemented in our Python framework it is almost as fast as the “ultrafast” MSFragger. In the benchmark, AlphaPept is still many times faster than the widely used MaxQuant despite using a similar search algorithm.

In terms of scientific novelty, AlphaPept is heavily based on the existing methods. The score function, X!Tandem, used in database searching has been proposed since 2003. AlphaPept uses the existing semi-supervised learning approach, which was proposed by Käll et al in 2007, to rescore PSMs. The label-free quantification was re-implemented based on MaxQuant (published in 2008). Furthermore, the speed-wise advantage over MaxQuant is also not due to the authors' novel contribution, rather, it is due to the usage of a popular python package, Numba (initially released in 2012).

We agree with the reviewer that AlphaPept implements several methods that were already described in the literature. However, often no code had been made available. In this respect, part of our contribution to the community is in overcoming this legacy gap: Not only do we make this code available but we also provide the respective framework in the form of Jupyter Notebooks and transparent performance evaluation – all in one, easily accessible programming language.

Programming is a continuous effort and functionality changes over time. Much of the speed of AlphaPept – despite being entirely written in Python – comes from the Numba JIT compiler. In Numba, we tried many implementations, some much slower than the final form. Many of the current design choices were only possible through recent updates to Numba and Python. Showing precisely how all this is done is fundamental for the broader proteomics community and for scientific progress.

We implemented X!Tandem and other scores to have well-established references score. However, based on the reviewers comments, in the latest version of AlphaPept, we now use a completely novel scoring function, which we call “Generic Score”. It is inspired by the Morpheus score (Number of Hits and matched intensity fraction), the intuition behind the X!Tandem hypergeometric distribution, and the length of the target like so:

$$hits \times \frac{hits}{2 \times n_frags_db} + hits \times fragments_matched_int_ratio + hits_y$$

In the revised manuscript, we test this score on 6 Bruker and Thermo files. It showed superior performance for Thermo data and slightly better performance for Bruker data while overall having a good overlap with X!Tandem and Morpheus Score (containing on average 98% of the X!Tandem and 97% of the Morpheus hits for Thermo (for Bruker 95 and 94%). The increase in identifications is 33% for Thermo and 6% for Bruker. Note that this does not increase

overall identifications as the ML-based score is still superior. More technical benchmarking of the generic score are now described in the text and Supplementary Material.

In the introduction section, unfortunately, the authors failed to describe the challenges in the bottom-up proteomics data analysis. As a result, I could not clearly get what scientific problems they wanted to address by developing this new software.

We thank the reviewer for pointing this out. We now extended the introduction with a paragraph to shed further light on the advantages of open-source software and testing. While in the past, results were publishable without including the respective code, this limits the transparency of the methods and somewhat defeats the purpose of publishing. In our opinion this is a fundamental problem for scientific methods. Current developments, also among publishers, increasingly mandate open access also to code and in some cases even require to maintain software and websites at least for some minimum time.

Besides the above issues, there are also several major and minor issues (see the following lists). I also have a small suggestion at the end, and hopefully, the authors could add the phosphoproteome and HLA data analysis if they have time.

We agree that the integration of phosphoproteome and HLA data analysis is of great interest, but we think it would be out of scope for this manuscript. We point the reviewer to our AlphaPeptDeep paper, which was published in Nature Communications in the meantime, and which has a state-of-the-art HLA data analysis pipeline (PMID 36433986).

We further point to DeepFLR, also just published in Nature Communications, which is a very promising, deep-learning-based approach to phosphoproteomics (PMID 37080984).

Major issues:

1. In the “peptide spectrum matching” section, the authors described that AlphaPept uses two stages of scoring: the first scoring only counts the matched fragments. Then, top-N peptides (N=10 by default) are used to calculate X!Tandem score. The reason for using fragment counting is for fast speed. And the reason for using X!Tandem score is to pick the most correct peptides. Since the first scoring uses a not-so-good score, there will be false negatives that are not in the top 10 candidates. To demonstrate that the two stages approach does not sacrifice the sensitivity too much, the authors should show the Venn diagram of identified PSMs between the two-stage approach and the one-stage approach (all peptides are scored using X!Tandem score).

We apologize for the lack of clarity in the manuscript, which we have now addressed. In our initial tests, we had of course increased N and found that Top-10 was more than sufficient to not have false negatives. In more detail: The initial fragment-counting step also calculates the matched fraction for each hit; the initial score is, therefore, the Morpheus score that was referenced in the paper. While this score is computationally simpler (X! has a factorial), it should not be inferior but, in fact superior. In the Morpheus publication, it was shown that, despite it being less complex to outperform X!Tandem. This is additionally confirmed by the test of our Generic Score.

To further address that our two-stage approach does not lose sensitivity due to the Top-N hits, we varied the number of Top-N hits over different scores. Note that the first score is still Morpheus, and we now vary the second score. However, as we increase the N in Top-N we should pick up potential false negatives that are not in the top 10 candidates. For the test dataset, we observed that the Morpheus is indeed better than X!Tandem and our new generic score in turn is better than Morpheus. Our ML models perform much better but are similar to each other regardless of the initial score to distinguish target / decoys. For larger Top-N values we see a slight decrease in performance for non-ML scores. We hypothesize that this is due to the increase of the decoy population. For our ML implementation, we do not see a clear trend for the number of Top-N. This is pointed out in the revised manuscript and the data are visualized in the new Suppl. Figure 1 (see just below).

2. In the runtime comparison of Figure 8c and 8f, did AlphaPept use GPU? If not, the authors should add the runtime with GPU since they emphasized the advantage in multiple places. If yes, the authors should add the runtime without GPU since it is the case for most users.

For the comparisons in Figure 8, we did not use GPU. We now specified that all runtimes are for CPU unless noted otherwise. We do agree with the reviewer that non-GPU is the case for most users. GPU processing for was indeed faster for some computational tasks (e.g., feature finding) on single files. However, especially when processing many files parallel processing with CPU is preferred. We have now extended the manuscript to highlight this.

Note that GPUs can achieve higher processing speeds over single CPUs and are therefore particularly useful for specialized operations, for instance, real-time-search. However, we found that for runs with default settings and especially when processing multiple files in parallel, CPUs with multiple processors are usually better suited.

3. The authors should upload the raw result files, parameter files, and log files of AlphaPept, MaxQuant, MSFragger, IonQuant, and FragPipe to Zenodo or figshare as required by the publication policy. It also makes the readers easier to evaluate and verify the experiment results.

We provide all code that was used to generate the manuscript figures as Jupyter Notebooks on our GitHub repository. We are also happy to upload a corresponding zip file to Nature Communications if desired. This also includes logs and parameters for respective experiments. For the 200 file and quantification benchmark, we used the publicly available datasets PXD015087 and PXD028735. In the revision, we uploaded the results, parameters, and logs for the benchmarking Figure 7 and Supplementary Table 1 to Zenodo: [10.5281/zenodo.10223453](https://zenodo.org/records/10223454). Reviewer access is made available through this link until publication.

<https://zenodo.org/records/10223454?token=eyJhbGciOiJIUzUxMiJ9.eyJpZCI6ImExNGE0MmVlTlViNGEtdmVlZmFmLTlyMDU5ODViMjczZCIsImRhdGEiOnt9LCJyYW5kb20iOiZlNTNmYzJINWE1Y2U5OTJhZTBjOTVhM2FmNzE5ZWl1MCJ9.pmtHEHSMGRdE6hWvBdtsdGMlpmFqJ2b0KvVkDytnX-IYvsU3wQxTElTCWUDAPtkUNwO1jFMnigbj4zsR7jF1PQ>

4. In Figure 8, I believe MSFragger can only perform database searching. I guess the FDR filtering and label-free quantification were performed by Philosopher and IonQuant in FragPipe. The authors should clarify the tools they used in the benchmarking. And also cite those tools' papers:

- Kong, A. T., Leprevost, F. V., Avtonomov, D. M., Mellacheruvu, D., & Nesvizhskii, A. I. (2017). MSFragger: ultrafast and comprehensive peptide identification in mass spectrometry-based proteomics. *Nature methods*, 14(5), 513-520.
- Teo, G. C., Polasky, D. A., Yu, F., & Nesvizhskii, A. I. (2020). Fast deisotoping algorithm and its implementation in the MSFragger search engine. *Journal of proteome research*, 20(1), 498-505.
- da Veiga Leprevost, F., Haynes, S. E., Avtonomov, D. M., Chang, H. Y., Shanmugam, A. K., Mellacheruvu, D., ... &

Nesvizhskii, A. I. (2020). Philosopher: a versatile toolkit for shotgun proteomics data analysis. *Nature methods*, 17(9), 869-870.

d. Yu, F., Haynes, S. E., Teo, G. C., Avtonomov, D. M., Polasky, D. A., & Nesvizhskii, A. I. (2020). Fast quantitative analysis of timsTOF PASEF data with MSFragger and IonQuant. *Molecular & Cellular Proteomics*, 19(9), 1575-1585.

e. Yu, F., Haynes, S. E., & Nesvizhskii, A. I. (2021). IonQuant enables accurate and sensitive label-free quantification with FDR-controlled match-between-runs. *Molecular & Cellular Proteomics*, 20.

We thank the reviewer for pointing this out. We have now added these citations.

5. In Figure 8, the authors should label “MSFragger+Philosopher+IonQuant” or “FragPipe” if they used those tools in the benchmarking. Also, correct the description in the main context.

We exchanged the label to FragPipe. and added the description in the main text.

Minor issues:

1. In Figure 3a, what does the “Numba, threaded” means? The authors should explain the meaning in the figure caption.

The explanation has been added to the figure caption:

CuPy refers to using GPU, Numba to single-threaded implementation, Numba threaded to Numba using multiple threads.

2. In line 283, there is no Fig 3d in the manuscript. Did the authors miss one figure?

Thank you. We have now updated the Figures and fixed the references

3. In line 341, the authors should cite two earlier publications:

a. Granholm, V., Navarro, J. F., Noble, W. S., & Käll, L. (2013). Determining the calibration of confidence estimation procedures for unique peptides in shotgun proteomics. *Journal of Proteomics*, 80, 123-131.

b. Feng, X. D., Li, L. W., Zhang, J. H., Zhu, Y. P., Chang, C., Shu, K. X., & Ma, J. (2017). Using the entrapment sequence method as a standard to evaluate key steps of proteomics data analysis process. *BMC Genomics*, 18(2), 1-9.

We have now added the references.

4. In line 355, Fig 4A is not about the target-decoy approach. I guess it should refer to Fig. 5A.

5. In line 360, it should refer to Fig. 5B, not Fig. 4B.

6. In line 370, it should refer to Figure 5c, not Figure 4c.

7. In line 373, it should refer to Fig. 5d, not 4d.

8. In line 577, it should refer to Fig. 9a, not 8a.

9. In line 602, it should refer to Fig. 9b, not 8b.

10. In line 615, it should refer to Fig. 9c, not 8c.

11. In line 617, it should refer to Fig. 9d, not 8d. The authors also forgot to label “9d” in the figure.

12. In line 671, it should refer to Table 2, not Table 1.

Thank you for picking this up. For the revised version, we have updated the figures and tables and have updated the references in the text.

Suggestion:

1. It would be great if the authors could demonstrate AlphaPept’s performance in analyzing phosphoproteome and HLA datasets.

We thank the author for this suggestion and point out that this is possible in principle with AlphaPept. However, as stated above we believe we could not do these themes justice in this manuscript and that there are other works potentially better for this kind of data and think this is out of scope for the current manuscript.

===

Reviewer #2 (Remarks to the Author):

Authors Strauss et al in their manuscript “AlphaPept, a modern and open framework for MS-based proteomics” revisit fundamental proteomics data processing with specific considerations for big data, cloud-based environments. In the provided demonstrations comparing AlphaPept to existing tools, AlphaPept does not typically outperform other search tools (particularly seems to underperform compared to the popular tool MSFragger), however AlphaPept is open source and written in Python, which is admittedly a major draw for some researchers in the community. After testing the software myself, I do think that AlphaPept will struggle to gain more users compared to MSFragger unless AlphaPept introduces some major draw for the average proteomicist. There are many other open source MS-based proteomics tools and frameworks – crux, comet, MSGF+, OpenMS – but I do not think it necessary to test every other software, and the two softwares tested here are probably the most popular used in the community for DDA.

Overall, the main novel aspect and noteworthy result of this work seems to be the language (Python) that it is written in.

We thank the author for the perspective on where AlphaPept stands and for sharing their thoughts about the principal challenge of where AlphaPept will be in the landscape of proteomics processing tools. We completely agree with their sentiment.

MINOR CONCERNS:

- In the timings table, it's odd to see that cloud-based searching takes so much longer than a local machine. The resources on the cloud-based search are more on-par with the laptop resources, which seems like a strange choice for demonstrating cloud-based deployment. I would like to see how AlphaPept works in a truly powerful cloud environment, as I imagine deployment on AWS or other cloud services would be a major use case for the type of users drawn to a tool like this. Could the authors repeat the cloud-based runs with better AWS resources, as a major advantage of their software is purportedly that can be deployed in a scalable cloud environment?

We thank the reviewer for bringing up the use of cloud resources. For a proteomics environment, we do see two major use cases: (1) Continuous processing of (all) incoming files and (2) processing of large studies. AlphaPept has the option to `continue_runs`, i.e., files processed with (1) can be re-used when running a full study (2). We refer to this in the manuscript text as using preprocessed files.

We chose a relatively small resource cloud setup because we aimed at use case (1) where only one file at a time is processed, in which case we obtain an estimate on the compute prices on a per-file basis. This would be a primary motivation for running in the cloud, as this scales with the number of computing resources available at AWS and can cover phases of high usage (e.g. running with a many samples per day method).

For use case (2), processing might only be needed for a specific study, and it might not be appropriate to buy a powerful processing PC for just this study. Instead, cloud computing would allow a more cost-effective access to powerful computing resources. In the revised manuscript, we also benchmarked this use case by repeating the study of 200 HeLa runs on two AWS cloud instances: m5.metal (96 cores with 3.1 GHz, 384 GB RAM) and c6i.32xlarge (128 cores, 3.5 GHz, 192 GB RAM) vs. our local processing station (24 cores with 3.5 GHz, 128 GB RAM).

Most notably, our local machine was the fastest (total time = 413 min), with the cloud instances 10-15% slower (total time = 444 min and 482 min for the 128 and 96 instances). We hypothesize that the major contributor to this is that most of the algorithms are disk-speed limited and not compute-limited. A breakdown of the individual processing steps shows indeed that the cloud instances were faster on compute-heavy tasks (such as search and score) but fell behind on raw import. We note that we ran this experiment on the faster IOPS store of AWS but apparently experienced varying disk speeds (i.e. Cloud 128 took 10x times longer for raw importing than Cloud 96) Note that, choosing the most expensive compute resources (on-demand), this would translate to about \$100 for a study of this kind. Again, when taking away steps that could be preprocessed in use case (1), such as import_raw_data and feature_finding, the total time would be about halved.

Since timings are mentioned throughout the text, we now moved the dedicated timing table to the Supplement.

- Colab notebook booted up fine for me, but was running slow and eventually crashed after using all available RAM. Could there be some comment added to the notebook describing how to allocate more RAM for this so that the demonstration completes?

We appreciate the reviewer testing the Colab notebook. Unfortunately, the resources Google provides for Colab can fluctuate and not changed from a user side. See the information below from Google.

Colab is able to provide resources free of charge in part by having dynamic usage limits that sometimes fluctuate, and by not providing guaranteed or unlimited resources. This means that overall usage limits as well as idle timeout periods, maximum VM lifetime, GPU types available, and other factors vary over time. Colab does not publish these limits, in part because they can (and sometimes do) vary quickly.

For the revision, we upgraded to the latest AlphaPept version, which should be less RAM intensive. When testing, the entire workflow took approx. 28 minutes to run through. We clarified the above points in the revised manuscript.

- Installation of AlphaPept with the Windows installer worked fine for me, but I found the tool a bit clunky, at least in the example vignette provided. The navigation is a bit clunky – for example, selecting which files to be run from a folder if there’s multiple files in the folder. I suppose that this is an argument for proper directory/subdirectory organization, but I can easily imagine situations where files from across directories are run and it would be a pain to copy-paste the files into multiple directories.

Our implementation indeed depends on proper directory/subdirectory organization, where we would see the researcher organizing all experimental data in one folder so that it can for example also subsequently be uploaded to PRIDE. However, based on the reviewer’s feedback, we have now updated the GUI and enabled the functionality to select and exclude files in a folder when multiple files are present. Attached is a screenshot of this functionality:

It would be nice to provide another vignette example that utilizes the other Results plots (Volcano plot, Correlation heatmap, PCA). Could the authors include a second vignette example dataset, perhaps something more typical of a DDA search, to demonstrate those plotting functions?

For the revised version, we have now changed the benchmark set to this publicly available one: PXD028735 (“A comprehensive Lfq benchmark dataset on modern day acquisition strategies in proteomics”). We have now also updated the documentation and show how these plotting functions can be used on this dataset. This includes results overview, volcano, correlation, scatter, pca-plots, and sequence coverage maps. These plots are meant for exploratory analysis. The new documentation can be found here: https://mannlabs.github.io/alphapept/proteomics_analysis.html

For further publication-ready analysis, we have since created a dedicated downstream package AlphaPeptStats (<https://github.com/MannLabs/alphapeptstats>), recently published in Bioinformatics (<https://doi.org/10.1093/bioinformatics/btad461>) that is also compatible with output formats from other search engines.

Reviewer #3 (Remarks to the Author):

Strauss et al. describe a processing pipeline for mass spectrometry-based proteomics data that they named AlphaPept. The pipeline is benchmarked to run fast, despite being mostly developed in python code, a feature that

the authors attribute to the numba python compiler. The pipeline is modularly built around a framework nbdev (<https://nbdev.fast.ai/>) which separates code and documentation from jupyter notebooks (Normally notebooks are great for visualization but tedious to program in). The nbdev allows for easy for inspection of the author's code but also allows the user to further add code in a straightforward manner.

The manuscript reads mostly as a walkthrough of the features of AlphaPept, and there is no actual novelty to the algorithmics. Despite its name, AlphaPept does not contain any deep learning. That said I am very pleased with the efforts. The contribution of the manuscript lies in that it describes the engineering behind a series of nice feature implementations and optimizations. Also, neither of the compared pipelines, MSFRagger and MaxQuant are open source or transparent to their inner workings. Hence this manuscript is a step forward compared to such pipelines. I am, however, aware of the current rise of a series of different pipeline implementations inside of the NextFlow environment, which to me seem as transparent and are likely easier to install and execute than AlphaPept. The rise of such pipelines should be mentioned in the introduction.

We thank the reviewer for lauding our efforts. The name AlphaPept was meant as a pun for “Alphabet” – the mother company of Google, with their open-source efforts in mind and was chosen before the increasing popularity of the deep learning packages such as AlphaGo and AlphaFold. That said our own recent deep learning efforts are in the AlphaPeptDeep package. Due to the inflationary appearance of AlphaX-packages that are from Google and deep learning based, we now see the limitations of this naming convention and will consider this for future packages.

In the revised manuscript, we have extended the introduction to mention Nextflow, which describes itself as a workflow manager (rather as a framework that one could put on top of AlphaPept). We also now provide docker containers for AlphaPept so it can be natively integrated.

We do not share the opinion of the reviewer regarding ease-of-use. To build a proteomics workflow with Nextflow, users must install at minimum Java, Nextflow, Docker, and the respective workflow. The easiest AlphaPept installation is the one-click installation on Windows, which is still the primary operating system for mass spec acquisition software. In contrast, the Nextflow installation on Windows is multiple pages long and consists of 9 core steps*. To give some further perspective, there was an insightful discussion about whether it makes sense to integrate FragPipe into Nextflow on GitHub**, which concluded with the developer deciding to integrate a headless mode – that AlphaPept also has, in favor of a Nextflow workflow for usability.

*<https://nextflow.io/blog/2021/setup-nextflow-on-windows.html>

**<https://github.com/Nesvilab/FragPipe/issues/518>

When investigating the running times in Table 2, I am missing a comparison point. As an academic researcher, my alternative to process data is not to pay for an AWS cloud. Instead, I use my university cluster which provides a SLURM queue, and I containerize my software using e.g. Singularity to assure smooth execution over 100s of cores in parallel. Please include a comparison to processing in such environments.

We hope we have been able to motivate the choice of hardware in response to reviewer #2 above. For a SLURM cluster, the above-mentioned use case (1) would allow smooth execution over all nodes but potentially poorly use the cluster as it will process one file per node, which will then be underutilized.

Use case (2) is bound to the hardware limits of each node. Here, a powerful workstation can exceed the performance of a node and is sometimes preferred by researchers.

Our SLURM cluster at Max Planck does not use Singularity; however, as AlphaPept is Python-based, we could use it on the cluster without further containerization. As we provide docker containers for AlphaPept they should be readily translatable to Singularity containers.

As the cluster runs on Linux we need mono to load Thermo files, which is currently limited to one process per node. This had total processing times of 1948 minutes without preprocessed files and 280 minutes with preprocessed files. This information is now included as Supplementary Table 1.

A major appeal of AWS is not only the access to almost unlimited and powerful hardware but also by the accompanying ecosystem. As an example, AWS has S3 buckets to store data and AWS batch to schedule workflows, which can be connected via triggers (lambda functions). To establish a workflow where every file that gets uploaded to the cloud is automatically processed is, therefore rather standard, whereas it would require some amount of scripting for local machines. The respective timings are now in Supplementary Table 1.

The benchmark against MSFragger and MaxQuant is apt, however, it is hard to get the conclusions from Figure 8. The ratio/sum scatter plots are hard to compare with the naked eye. Add a direct comparison of the number of reported differentially abundant peptides as a function of FDR for the three methods. Also, you have to spell out what is meant by “quantified” peptides and proteins in the Venn diagrams of Figures 8b&8e, e.g. are they above a FDR threshold? A FC Treshold?

We thank the reviewer for pointing out the limitations in the benchmark of Figure 8. We now decided to switch to benchmarking on a dataset that was created for such a comparison, “A comprehensive LFQ benchmark dataset on modern day acquisition strategies in proteomics” from Van Puyvelde et al. (PMID 35354825). We agree that the ratio/scatter plots can be hard to interpret and provide rather qualitatively insight. However, they were also used in the benchmark study of that paper.

For a more quantitative metric we have now created Supplementary Figure 4 and 5 showing the delta to the true ratio for each species over the number of proteins. In the experiment, we expect an actual fold change based on the spike-in ratios when the sample was pipetted. We can therefore calculate a delta to this true ratio and compare this to the number of proteins that were identified. Performance-wise, AlphaPept is in between MSFragger (better) and MaxQuant (worse).

We note that the idea of this benchmark is to show the entire pipeline, i.e., from raw data to quantified proteins as it is difficult to find comparable settings for individual steps. A comparison on peptide-level would not cover parts of the downstream algorithms to determine protein intensities. Additionally, when investigating at the peptide-level, for the comparison of the number of reported differentially abundant peptides as a function of FDR, this could potentially be misleading. For one, this can only be meaningfully computed for the class that changes abundance, and a method with low accuracy and high precision but shifted to the right would look better than the same method with high accuracy and high precision.

We hope this this explanation satisfies the reviewer. To nonetheless report on peptide identification, we investigated the number of Arabidopsis hits as a measure of the FDR. When searching the benchmark set, we had additionally added Arabidopsis FASTA as an external validation. We therefore now included a Supplementary Figure 6 and 7 showing the number of identified peptides versus the Arabidopsis FDR. Here, AlphaPept seems to be more conservative for the top ranked identifications compared to the other software packages while never exceeding the 1% FDR.

We designate ‘quantified proteins’ as the proteins that are still present after LFQ optimization. This constitutes a minimum number of peptide occurrences across the runs to calculate a relative protein intensity.

REVIEWERS' COMMENTS

Reviewer #1 (Remarks to the Author):

Please see the attached file.

Reviewer #1 (Remarks on code availability):

I reviewed and tested the code. There is README with very informative details and instructions for installing and running the program. I was able to install and run the code.

Reviewer #2 (Remarks to the Author):

The authors have satisfactorily addressed my previous concerns, thank you for these efforts,

Two minor new concerns:

- I noted that the zenodo link was not available, even the "reviewer link" was not visible.

- In the response to reviewers figure titled "Compute times 200 proteomes - Cloud vs Local" -- are the axes swapped? I think "step" and "Time (min, logscale)" labels are on the wrong axes!

Thank you for addressing my concerns and I look forward to seeing this work published.

Reviewer #3 (Remarks to the Author):

The revised manuscript reads better. However, I still see a problematic point with it. I have attached the relevant parts from my comment and reply:

>> I am, however, aware of the current rise of a series of different pipeline implementations inside of the NextFlow environment, which to me seem as transparent and are likely easier to install and execute than AlphaPept. The rise of such pipelines should be mentioned in the introduction.

> We do not share the opinion of the reviewer regarding ease-of-use. To build a proteomics workflow with Nextflow, users must install at minimum Java, Nextflow, Docker, and the respective workflow. The easiest AlphaPept installation is the one-click installation on Windows, which is still the primary operating system for mass spec acquisition software. In contrast, the Nextflow installation on Windows is multiple pages long and consists of 9 core steps*. To give some further perspective, there was an insightful discussion about whether it makes sense to integrate FragPipe into Nextflow on GitHub**, which concluded with the developer deciding to integrate a headless mode – that AlphaPept also has, in favor of a Nextflow workflow for usability.

*<https://nextflow.io/blog/2021/setup-nextflow-on-windows.html>

**<https://github.com/Nesvilab/FragPipe/issues/518>

Here, I should emphasize that I do not have any vested interests in Nextflow, or nf-core, but still, I must insist that the installation of Nextflow is very straightforward. Particularly the transition between execution on a local workstation and clusters is extremely easy in their system. I have successfully been using the quantms pipeline without any difficulties, either in installation or execution, for the last year or so. While the authors are right in that most mass spectrometrists still use Windows, the most serious data processing efforts are made under Linux-based systems. The migration towards Linux and cluster-based processing is an accelerating trend, much dependent on the increasing sizes of the data files for each new generation of mass spectrometrists. This development should be acknowledged, at least peripheral in the manuscript, e.g. in the Introduction or the Discussion.

In this revision, the authors have addressed most of my comments. Only a few minor issues remain. I also have comments on some of the responses. Please find those in the following. For the authors' other responses, I didn't include them here to make the letter short.

Authors' original response: For the revision, we have updated parts of the code and provided a more extensive benchmark set. In these measurements, FragPipe-MSFragger is still the fastest in class. We speculate that this is due to the conceptually different search algorithm being used, which has an edge in identification, especially for open searches. However, we note that AlphaPept is modular and the algorithm can in principle be replaced, for instance by the MSFragger algorithm. The current search algorithm is a rather conventional fragment-counting algorithm. Implemented in our Python framework it is almost as fast as the "ultrafast" MSFragger. In the benchmark, AlphaPept is still many times faster than the widely used MaxQuant despite using a similar search algorithm.

Reviewer 1's comment: Yes, I agree that AlphaPept and MSFragger use different algorithms. I would also like to point out that the programming language might also significantly affect the speed. Although AlphaPept has used several advanced Python packages and tricks to mitigate Python's slow-speed issue, it is still not the same as using languages, such as Java.

Authors' original response: Programming is a continuous effort and functionality changes over time. Much of the speed of AlphaPept – despite being entirely written in Python – comes from the Numba JIT compiler. In Numba, we tried many implementations, some much slower than the final form. Many of the current design choices were only possible through recent updates to Numba and Python. Showing precisely how all this is done is fundamental for the broader proteomics community and for scientific progress.

Reviewer 1's comment: Thank you for the additional information. I agree that writing a fast and robust program is not trivial, and requires significant effort.

Authors' original response: We implemented X!Tandem and other scores to have well-established references score. However, based on the reviewers comments, in the latest version of AlphaPept, we now use a completely novel scoring function, which we call "Generic Score". It is inspired by the Morpheus score (Number of Hits and matched intensity fraction), the intuition behind the X!tandem hypergeometric distribution, and the length of the target like so: In the revised manuscript, we test this score on 6 Bruker and Thermo files. It showed superior performance for Thermo data and slightly better performance for Bruker data while overall having a good overlap with X!Tandem and Morpheus Score (containing on average 98% of the X!Tandem and 97% of the Morpheus hits for Thermo (for Bruker 95 and 94%). The increase in identifications is 33% for Thermo and 6% for Bruker. Note that this does not increase overall identifications as the ML-based score is still superior. More technical benchmarking of the generic score are now described in the text and Supplementary Material.

Reviewer 1's comment: Thanks for the updated information. It is a little counterintuitive that the Morpheus score is better than the X!Tandem score because the X!Tandem score uses more information regarding the different ion types (i.e., separated terms for N-term and C-term ions) and intensities. Are the Venn diagrams in Supplementary Figure 2 and 3

only from closed search? If yes, I would suggest adding the comparisons using open search which has a much larger search space.

Authors' original response: We apologize for the lack of clarity in the manuscript, which we have now addressed. In our initial tests, we had of course increased N and found that Top-10 was more than sufficient to not have false negatives. In more detail: The initial fragment-counting step also calculates the matched fraction for each hit; the initial score is, therefore, the Morpheus score that was referenced in the paper. While this score is computationally simpler ($X!$ has a factorial), it should not be inferior but, in fact superior. In the Morpheus publication, it was shown that, despite it being less complex to outperform $X!$ Tandem. This is additionally confirmed by the test of our Generic Score. To further address that our two-stage approach does not lose sensitivity due to the Top-N hits, we varied the number of Top-N hits over different scores. Note that the first score is still Morpheus, and we now vary the second score. However, as we increase the N in Top-N we should pick up potential false negatives that are not in the top 10 candidates. For the test dataset, we observed that the Morpheus is indeed better than $X!$ Tandem and our new generic score in turn is better than Morpheus. Our ML models perform much better but are similar to each other regardless of the initial score to distinguish target / decoys. For larger Top-N values we see a slight decrease in performance for non-ML scores. We hypothesize that this is due to the increase of the decoy population. For our ML implementation, we do not see a clear trend for the number of Top-N. This is pointed out in the revised manuscript and the data are visualized in the new Suppl. Figure 1 (see just below).

Reviewer 1's comment: Thanks for the clarification. I am a little confused by several small details,

- 1. What is the score function in the initial fragment-counting step? From the manuscript, it seems to be the number of matched fragment ions without considering intensity. However, you said, "the initial fragment-counting step also calculates the matched fraction for each hit", which confused me. Could you please clarify this and add the details to the manuscript so that the other readers will not be confused.**
- 2. In the PSMs list for the machine learning rescoring, how many candidates are there for each scan (1 or 10)?**

Authors' original response: For the comparisons in Figure 8, we did not use GPU. We now specified that all runtimes are for CPU unless noted otherwise. We do agree with the reviewer that non-GPU is the case for most users. GPU processing for was indeed faster for some computational tasks (e.g., feature finding) on single files. However, especially when processing many files parallel processing with CPU is preferred. We have now extended the manuscript to highlight this. Note that GPUs can achieve higher processing speeds over single CPUs and are therefore particularly useful for specialized operations, for instance, real-time-search. However, we found that for runs with default settings and especially when processing multiple files in parallel, CPUs with multiple processors are usually better suited.

Reviewer 1's comment: Thanks for the updated information. It makes sense that CPU is more suitable for tasks such as database searching because, most of the time, IO, not computing, is the bottleneck.

Reviewer 1's original comment: In Figure 8, the authors should label "MSFragger+Philosopher+IonQuant" or "FragPipe" if they used those tools in the benchmarking. Also, correct the description in the main context.

Authors' original response: We exchanged the label to FragPipe. and added the description in the main text.

Reviewer 1's comment: There are still "MSFragger" and "MSFragger LFQ" in Figure 7 (old Figure 8) a and d, which does not seem correct because MSFragger is a search engine. Those quantitative data should be from the whole workflow in FragPipe. Please change these to "FragPipe Intensity" and "FragPipe MaxLFQ Intensity".

REVIEWER COMMENTS – FINAL

AlphaPept: a modern and open framework for MS-based proteomics

Reviewer #1 (Remarks to the Author):

For Reviewer #1, we only copied sections with questions from the attached file to keep this document short.

Thanks for the updated information. It is a little counterintuitive that the Morpheus score is better than the X!Tandem score because the X!Tandem score uses more information regarding the different ion types (i.e., separated terms for N-term and C-term ions) and intensities. Are the Venn diagrams in Supplementary Figure 2 and 3 only from closed search? If yes, I would suggest adding the comparisons using open search, which has a much larger search space.

We agree with the reviewer that the performance of the Morpheus score is remarkable, which was also noted in the initial Morpheus publication. It was then postulated that “the added specificity of high-mass accuracy data makes it easier to distinguish correct and incorrect identifications with a less finely tuned metric.”

Supplementary Figures 2 and 3 results were from a closed search with a precursor tolerance of 50 ppm. While one can adjust the precursor tolerance with AlphaPept, an entirely open search over hundreds of Daltons, as in MSFragger, would probably take quite long due to conceptual difference in search algorithms.

However, to simulate the effect of an increasing precursor tolerance, we varied the tolerance from 5 ppm to 6400 ppm (i.e. 6.4 Da at a mass of 1000). We searched each of one file of the previous test sets and compared the scores: Generic – our new generic score, x_tandem for X!Tandem, default for Morpheus and baseline for the number of shared precursors between the three scores.

Overall, the curves show non-linear effects, with our generic score giving the best performance across the parameter space. For the comparison between X!Tandem and Morpheus performs better in the range of approx. 50 to 500 ppm

for Thermo data, and up to 1000 ppm for Bruker data. In the Bruker test case, this trend reverses at approx.. 4500 ppm.

We speculate that there is a complex interplay of precursor tolerance, raw file calibration, score information content, and target decoy competition. While our initial observation would still hold true for the tested precursor tolerance, we think further analysis would be necessary. As this would be out of scope for the manuscript, we have since removed the sentence comparing X!Tandem and Morpheus from the main text.

~~In our benchmarks it showed superior performance over X!Tandem and Morpheus (Supplementary Figs. 2, 3).~~

Reviewer 1's comment: Thanks for the clarification. I am a little confused by several small details,

1. What is the score function in the initial fragment-counting step? From the manuscript, it seems to be the number of matched fragment ions without considering intensity. However, you said, "the initial fragment-counting step also calculates the matched fraction for each hit", which confused me. Could you please clarify this and add the details to the manuscript so that the other readers will not be confused.

We thank the reviewer for pointing this out. We have now revised this paragraph for clarity:

To achieve maximum speed, AlphaPept employs a very rapid fragment-counting step to determine initial peptide spectrum matches (PSMs). In brief, this step involves two pointers that iterate over the lists of sorted theoretical and experimental fragment masses and compare their mass difference. If the mass difference between a fragment pair falls within the search tolerance, this is considered a hit, and the counter is incremented by one. As this step only involves addition and subtraction of elements in numerical arrays, the machine code produced by Numba is very efficient and easily parallelized. For each hit, we additionally compute the fraction of MS2 signals accounted for by the match and add this fraction to the counter. Consequently, for each peptide-spectrum match, we store a floating-point number that represents the integer number of hits and the matched intensity fraction of these hits, effectively re-implementing the Morpheus score. This score, despite its computational simplicity can outperform the somewhat more complex X!Tandem score (Wenger and Coon 2013). This leaves a much smaller number of peptides that have at least a minimum number of fragment matches to the experimental spectrum.

2. In the PSMs list for the machine learning rescoring, how many candidates are there for each scan (1 or 10)?

The default Top-N setting allows for ten potential candidates for each query. However, the number of candidates can vary, as each scan may encompass multiple queries. Specifically, when aligning MS1 features with MS2 spectra, each feature and scan can have several assignments. Additionally, there is a minimum score threshold of min_frag_hits (defaulting to 7) for fragment hits, so that not all 10 candidates are assigned. Therefore, a scan with a single feature may have up to 10 Peptide-Spectrum Match (PSM) candidates. For scans with multiple features, the number of candidates increases accordingly. In our scoring process, we rank the features while ensuring that each feature is matched only once.

There are still "MSFragger" and "MSFragger LFQ" in Figure 7 (old Figure 8) a and d, which does not seem correct because MSFragger is a search engine. Those quantitative data should be from the whole workflow in FragPipe. Please change these to "FragPipe Intensity" and "FragPipe MaxLFQ Intensity".

We changed the Figure accordingly.

Reviewer #1 (Remarks on code availability):

I reviewed and tested the code. There is README with very informative details and instructions for installing and running the program. I was able to install and run the code.

We thank the reviewer for testing the code.

Reviewer #2 (Remarks to the Author):

The authors have satisfactorily addressed my previous concerns, thank you for these efforts,

Two minor new concerns:

- I noted that the zenodo link was not available, even the "reviewer link" was not visible.

We thank the reviewer for picking this up. We now checked and saw that the reviewer link was quite long and line breaks might have caused issues when clicking the link. We have now used a URL shortening service to make it better accessible (<http://tinyurl.com/ZENODOAP>) and verified that the Zenodo record number (10223454) exists <https://zenodo.org/records/10223454>. At the point of writing, the entry was viewed 30 times.

- In the response to reviewers figure titled "Compute times 200 proteomes - Cloud vs Local" -- are the axes swapped? I think "step" and "Time (min, logscale)" labels are on the wrong axes!

We thank the reviewer for pointing this out. The axes were indeed swapped. Attached is the revised Figure:

Thank you for addressing my concerns and I look forward to seeing this work published.

We thank the reviewer for supporting our work.

Reviewer #3 (Remarks to the Author):

The revised manuscript reads better. However, I still see a problematic point with it. I have attached the relevant parts from my comment and reply:

>> I am, however, aware of the current rise of a series of different pipeline implementations inside of the NextFlow environment, which to me seem as transparent and are likely easier to install and execute than AlphaPept. The rise of such pipelines should be mentioned in the introduction.

> We do not share the opinion of the reviewer regarding ease-of-use. To build a proteomics workflow with Nextflow, users must install at minimum Java, Nextflow, Docker, and the respective workflow. The easiest AlphaPept

installation is the one-click installation on Windows, which is still the primary operating system for mass spec acquisition software. In contrast, the Nextflow installation on Windows is multiple pages long and consists of 9 core steps*. To give some further perspective, there was an insightful discussion about whether it makes sense to integrate FragPipe into Nextflow on GitHub**, which concluded with the developer deciding to integrate a headless mode – that AlphaPept also has, in favor of a Nextflow workflow for usability.

*<https://nextflow.io/blog/2021/setup-nextflow-on-windows.html> **<https://github.com/Nesvilab/FragPipe/issues/518>

Here, I should emphasize that I do not have any vested interests in Nextflow, or nf-core, but still, I must insist that the installation of Nextflow is very straightforward. Particularly the transition between execution on a local workstation and clusters is extremely easy in their system. I have successfully been using the quantms pipeline without any difficulties, either in installation or execution, for the last year or so. While the authors are right in that most mass spectrometrists still use Windows, the most serious data processing efforts are made under Linux-based systems. The migration towards Linux and cluster-based processing is an accelerating trend, much dependent on the increasing sizes of the data files for each new generation of mass spectrometrists. This development should be acknowledged, at least peripheral in the manuscript, e.g. in the Introduction or the Discussion.

We thank the reviewer for sharing his experience with quantms. This indeed sounds very promising and we will be testing the Linux version in the future. We had previously mentioned the Nextflow framework in the Introduction. This was now extended to include the aspect of scalable execution and containerization. We now also mention nf-core and quantms.

There are increased efforts to achieve reproducible computational workflows by using pipelines such as the Nextflow framework (Di Tommaso et al. 2017) that allow scalable execution using software containers. Community efforts such as nf-core (Ewels et al. 2020) build on Nextflow for collaborative, peer-reviewed, best-practice pipelines, such as quantms for proteomics (Yasset Perez-Riverol et al. 2023).